# GAUSSIAN PROCESS META-REPRESENTATIONS FOR NEURAL NETWORKS

## ABSTRACT

Bayesian inference offers a theoretically grounded and general way to train neural networks and can potentially give calibrated uncertainty. It is, however, challenging to specify a meaningful and tractable prior over the network parameters. More crucially, many existing inference methods assume mean-field approximate posteriors, ignoring interactions between parameters in high-dimensional weight space. To this end, this paper introduces two innovations: (i) a Gaussian process-based hierarchical model for the network parameters based on recently introduced unit embeddings that can flexibly encode weight structures, and (ii) input-dependent contextual variables for the weight prior that can provide convenient ways to regularize the function space being modeled by the network through the use of kernels. Furthermore, we develop an efficient structured variational inference scheme that alleviates the need to perform inference in the weight space whilst retaining and learning non-trivial correlations between network parameters. We show these models provide desirable test-time uncertainty estimates, demonstrate cases of modeling inductive biases for neural networks with kernels and demonstrate competitive predictive performance of the proposed model and algorithm over alternative approaches on a range of classification and active learning tasks.

## 1 INTRODUCTION

Bayesian neural networks (BNNs) (see e.g. MacKay, 1992; Neal, 1993; Ghahramani, 2016) are one of the research frontiers on combining Bayesian inference and deep learning, potentially offering flexible modelling power with calibrated predictive performance. In essence, applying probabilistic inference to neural networks allows all plausible network parameters, not just the most likely, to be used for predictions. Despite some earlier success in supervised learning tasks (Guyon et al., 2005; Quiñonero-Candela et al., 2006), BNNs has enjoyed limited recent practical applications. This could be attributed to: (i) the need to choose an appropriate weight prior as commonly modelers care about *functions priors* rather than *weight priors* but require the power and expressivity of weight-based models, and (ii) recently developed approximate inference methods are not able to capture complicated posterior correlations in large networks. As such, addressing these challenges is an active research area.

This paper attempts to remedy the aforementioned limitations for Bayesian neural networks by extending previous work on unit-based representations for neural networks with a unit-based Gaussian process hierarchical prior over network parameters, which flexibly models the correlations between weights in a layer and across layers while forgoing the need for a parametric hyper-network. Moreover, input-dependent contextual variables are used for modulating the GP-function and in turn specifying input-dependent weight priors which draw from the tools from the kernel literature to imbue neural networks with inductive biases. This may yield more desirable uncertainty estimates at test time, or restrict function spaces to generalize in particular ways. A structured variational inference approach is employed that side-steps the need to do inference in the weight space whilst retaining weight correlations in the approximate posterior. These theoretical benefits translate to competitive practical performance on a range of supervised and active learning tasks. The paper is organized as follows: in the next section, we will briefly review mean-field variational inference for BNNs and a recently proposed hierarchical model for BNNs. The proposed model and inference scheme are discussed in section 3 and section 4, respectively, followed by a discussion on related work in section 5. We then validate the performance of the proposed model and algorithm in section 6.

## 2 BACKGROUND

In this section, we lay out the groundwork for the paper by providing a concise summary of variational approaches to Bayesian neural networks and a recent approach to learning meta-representations of neural networks.

### 2.1 BAYESIAN NEURAL NETWORKS AND VARIATIONAL INFERENCE

Consider a training set comprising of $N$ input-output pairs, $\mathcal{D} = \{\mathbf{x}_n, y_n\}_{n=1}^N$, and a neural network parameterized by weights and biases, $\mathbf{w}$, that describes the distribution over an output $y_n$ given an input $\mathbf{x}_n$, $p(y_n|\mathbf{w}, \mathbf{x}_n)$. We follow a Bayesian approach by placing a prior distribution over the network parameters, $p(\mathbf{w})$, and obtaining the posterior distribution $p(\mathbf{w}|\mathcal{D})$, which involves calulation of the marginal likelihood $p(\mathcal{D}) = \int d\mathbf{w} p(\mathbf{w}) p(\mathcal{D}|\mathbf{w})$. However, obtaining $p(\mathbf{w}|\mathcal{D})$ and $p(\mathcal{D})$ exactly is intractable when $N$ is large or when the network is large and as such, approximation methods are often required. In particular, mean-field Gaussian variational inference (MFVI) has recently become a method of choice for approximate inference for Bayesian neural networks due to its simplicity and the recently popularized *reparameterization trick* (Salimans & Knowles, 2013; Kingma & Welling, 2013; Titsias & Lázaro-Gredilla, 2014; Blundell et al., 2015). MFVI sidesteps the intractability by positing a diagonal Gaussian approximation $q(\mathbf{w}) = \mathcal{N}(\mathbf{w}; \boldsymbol{\mu}, \mathrm{diag}(\boldsymbol{\sigma}^2))$ and optimising an approximate lower bound to the marginal likelihood $\mathcal{L}_{\mathrm{MFVI}}(q(\mathbf{w})) \approx -\mathrm{KL}[q(\mathbf{w})||p(\mathbf{w})] + \frac{1}{K} \sum_{k=1}^K \sum_{n=1}^N \log p(y_n|\mathbf{w}_k, \mathbf{x}_n)$, where $\mathbf{w}_k = \boldsymbol{\mu} + \boldsymbol{\sigma} \odot \epsilon_k$ and $\epsilon_k \sim \mathcal{N}(\mathbf{0}, \mathbf{I})$, i.e. $\mathbf{w}_k$ is a sample from $q(\mathbf{w})$. Note that the mean-field variational Gaussian approximation with a standard normal prior, presented in is often outperformed by point estimation in certain settings (Trippe & Turner, 2018). Despite being practical and able to give reasonable uncertainty estimates, improving MFVI is still an active research area, and the main focuses of which are (i) improving the reparameterization gradient estimator to enable faster convergence (Miller et al., 2017; Wu et al., 2018), (ii) replacing the typical standard Normal prior, $p(\mathbf{w}) = \mathcal{N}(\mathbf{w}; \mathbf{0}, \mathbf{I})$ by a structured prior that better models the structures present in the weight a-priori (Ghosh et al., 2018; Neal, 2012; Blundell et al., 2015), and (iii) using structured variational approximations that can potentially capture weight correlations in the posterior (Louizos & Welling, 2016; Zhang et al., 2017). This paper builds on the two latter themes and proposes a hierarchical model for the prior and a structured variational scheme that explicitly model and infer weight structures.

### 2.2 META-REPRESENTING WEIGHTS AND NETWORKS

Our work builds on and expands the class of hierarchical neural network models based on the concept of latent variables associated with units in a network as proposed in Karaletsos et al. (2018). In these models, each unit (visible or hidden) of the $l$-th layer of the network has a corresponding latent hierarchical variable $\mathbf{z}_{l,i}$, possibly of high dimensions $D_z$, where $i$ denotes the index of the unit in a layer. Note that these latent variables do not describe the activation of units, but rather constitute latent features associated with a unit. These latent variables are then used to construct the weights in the network. Their construction is judiciously chosen such that a weight in the $l$-th weight layer, $w_{l,i,j}$ is linked to the latent variables $z$'s of the $i$-th input unit and the $j$-th output unit of the weight layer. We can summarize this relationship by introducing the concept of *weight codes* $\mathbf{C}_w(\mathbf{z})$ consisting of the set of weight encodings for each individual weight in the network $\mathbf{c}_{w_{l,i,j}} = [\mathbf{z}_{l+1,i}, \mathbf{z}_{l,j}]$, which can be deterministically constructed from the collection of unit latent variables $\mathbf{z}$ by concatenating them correctly. Mathematically, the probabilistic description of the relationship between the weight codes (summarizing the structured latent variables) and the weights $\mathbf{w}$ is:

$$p(\mathbf{w}|\mathbf{z}) = p(\mathbf{w}|\mathbf{C}_w(\mathbf{z})) = \prod_{l=1}^{L-1} \prod_{i=1}^{H_l} \prod_{j=1}^{H_{l+1}} p(w_{l,i,j}|\mathbf{z}_{l+1,i}, \mathbf{z}_{l,j}), \tag{1}$$

where $l$ denotes a visible or hidden layer and $H_l$ is the number of units in that layer, and $\mathbf{w}$ denotes all the weights in this network.

The central motivations for this parameterization are three-fold. *First*, weights in a layer and across layers are explicitly correlated in the modelling stage, as they share latent variables if they are connected to the same unit, unlike the standard normal prior typically used as described in the last section. *Second*, the number of visible and hidden units in a neural network is typically much smaller than the number of weights. For example, for the $l$-th weight layer, there are $H_l \times H_{l+1}$ weights compared to $H_l + H_{l+1}$ associated latent variables. This encourages the development of inference

schemes that can work in the lower-dimensional hierarchical latent space directly without the need to infer the weights. *Third*, as hierarchical latent variables are associated with units in the network and have the same cardinalities, inferring these latent variables conditioning on observed data could be thought of as a form of network representation learning.

In Karaletsos et al. (2018), a small parametric neural network regression model (conceptually a *structured hyper-network*) is chosen to map the latent variables to the weights. The network either assumes a Gaussian noise model which factorizes over weights, $p(w_{l,i,j}|\mathbf{z}_{l+1,i}, \mathbf{z}_{l,j}, \theta) = \mathcal{N}(w_{l,i,j}; \mu_{l,i,j}, \sigma_{l,i,j}^2)$, where $(\mu_{l,i,j}, \log \sigma_{l,i,j}) = \text{NN}_\theta([\mathbf{z}_{l+1,i}, \mathbf{z}_{l,j}])$, or models dependence between weights implicitly as $p(\mathbf{w}|\mathbf{C}_w(\mathbf{z}), \theta) \propto \int p(\epsilon) \prod_{l,i,j} \text{NN}_\theta([\mathbf{z}_{l+1,i}, \mathbf{z}_{l,j}, \epsilon]) d\epsilon$, where $\epsilon$ is a random variate, $w_{l,i,j} = \text{NN}_\theta([\mathbf{z}_{l+1,i}, \mathbf{z}_{l,j}, \epsilon])$, and $\theta$ denotes the parameters of the neural network mapping. We will call this network a *meta mapping*. Note that given the collection of sampled unit variables $\mathbf{z}$, the weights (or theirs mean and variance) can be obtained efficiently in parallel. A prior over the latent variables $\mathbf{z}$, $p(\mathbf{z}) = \mathcal{N}(\mathbf{z}; \mathbf{0}, \mathbf{I})$, completes the model specification. We can thus write down the joint density of the resulting hierarchical model as follows,

$$p(\mathbf{y}, \mathbf{w}, \mathbf{z}|\mathbf{x}, \theta) = \left[\prod_{l=1}^{L} p(\mathbf{z}_l)\right] [p(\mathbf{w}|\mathbf{C}_w(\mathbf{z}), \theta)] \left[\prod_{n=1}^{N} p(\mathbf{y}_n|\mathbf{w}, \mathbf{x}_n)\right]. \tag{2}$$

Variational inference was employed in prior work to infer $\mathbf{z}$ (and $\mathbf{w}$ implicitly), and to obtain a point estimate of $\theta$, as a by-product of optimising the variational lower bound.

## 3 META-REPRESENTATIONS WITH GAUSSIAN PROCESS MAPPINGS AND CONTEXTUAL LATENT VARIABLES

In this section, we present two novel extensions of the hierarchical model in section 2.2 that aim to increase the robustness in the small data settings and improve its out-of-sample uncertainty estimates.

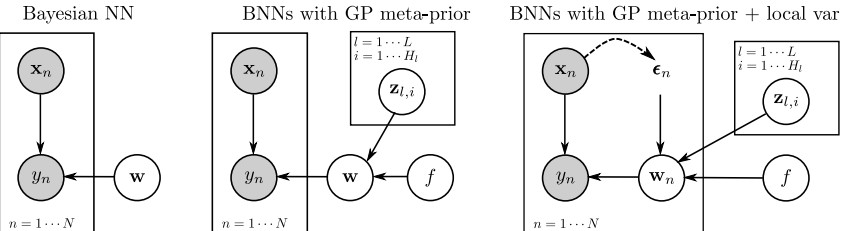

Figure 1: Graphical depiction of various models: vanilla BNNs, BNNs with hierarchical GP-MetaPriors, and BNNs with hierarchical GP-MetaPriors and auxiliary variables.

### 3.1 META-REPRESENTING WEIGHTS WITH GAUSSIAN PROCESSES

Notice that in section 2.2, the meta mapping from the hierarchical latent variables to the weights is a parametric non-linear function, specified by a neural network. We replace the parametric neural network by a probabilistic functional mapping and place a nonparametric Gaussian process (GP) prior over this function. That is,

$$p(w_{l,i,j}|f, \mathbf{c}_{w_{l,i,j}}) = \mathcal{N}(w_{l,i,j}; f([\mathbf{z}_{l+1,i}, \mathbf{z}_{l,j}]), \sigma_w^2);$$
$$p(f|\gamma) = \mathcal{GP}(f; \mathbf{0}, k_{c_w}(\cdot, \cdot|\gamma)), \tag{3}$$

where we have assumed a zero-mean GP, $k_\gamma(\cdot, \cdot)$ is a covariance function and $\gamma$ is a small set of hyper-parameters. The representational effect is that the latent function introduces correlations for the individual weight predictions,

$$P(\mathbf{w}|\mathbf{z}) = P(\mathbf{w}|\mathbf{C}_w(\mathbf{z})) = \int p(f) \Big[\prod_{l=1}^{L-1} \prod_{i=1}^{H_{l+1}} \prod_{j=1}^{H_l} p(w_{l,i,j}|f, \mathbf{z}_{l+1,i}, \mathbf{z}_{l,j})\Big] df. \tag{4}$$

Here, we present a homoscedastic noise model for the weights, but the model is readily adaptable to a heteroscedastic noise model which we omit for clarity. For a more detailed discussion on GPs and their applications, please see Rasmussen & Williams (2005). It is worth noting that whilst the number of latent variables and weights can be large, the input dimension to the GP mapping is only $2D_z$, where $D_z$ is the dimensionality of each latent variable $\mathbf{z}$. The GP mapping effectively performs one-dimensional regression from latent variables to individual weights while capturing their correlations. We will refer to this mapping as a **GP-MetaPrior** (*metaGP*).

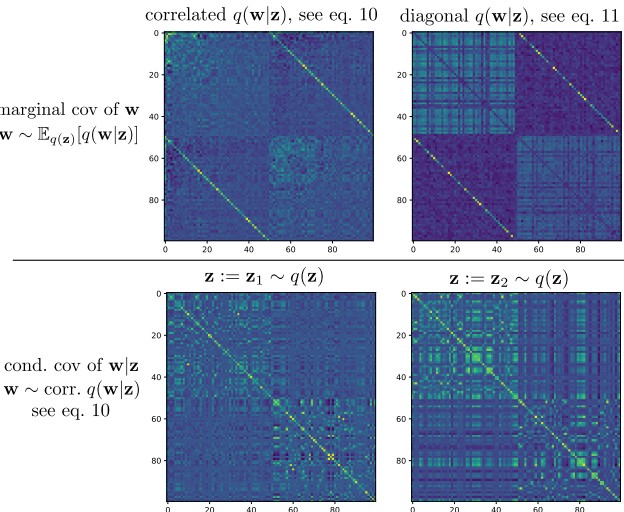

Figure 2: Marginal and conditional covariance structures over weights in a 1x50x1 neural network. Sampling from the posterior of the hierarchical model reveals that even a diagonal GP approximation can capture off-diagonal correlations induced through unit correlations. Also note the off-diagonal bands in the marginal plots above, which indicate the correlation structures induced by the latent variables of the hidden units connecting the layers.

In general, the form of the covariance function and the values of the hyperparameters encapsulate prior knowledge about the unknown function. We note that GPs benefit from the fact that kernels are compositional and can thus be used to represent correlation structures explicitly. We exploit this fact and define the following factorized kernel at the example of two weights in the network,

$$
\begin{aligned}
k_{c_w}(c_{w^1}, c_{w^2}) &= k([\mathbf{z}_{l^1+1,i^1}, \mathbf{z}_{l^1,j^1}], [\mathbf{z}_{l^2+1,i^2}, \mathbf{z}_{l^2,j^2}]) \\
&= k_{out}(\mathbf{z}_{l^1+1,i^1}, \mathbf{z}_{l^2+1,i^2}) \cdot k_{in}(\mathbf{z}_{l^1,j^1}, \mathbf{z}_{l^2,j^2}).
\end{aligned}
\tag{5}
$$

In this section and what follows, we will use the popular exponentiated quadratic kernel with ARD lengthscales, $k(\mathbf{x}_1, \mathbf{x}_2) = \sigma_k^2 \exp\left(\sum_{d=1}^{2D_z} \frac{-(x_{1,d} - x_{2,d})^2}{2l_d^2}\right)$, where $\{l_d\}_{d=1}^{2D_z}$ are the kernel lengthscales and $\sigma_k^2$ is the kernel variance. For the model considered here, this admits the following separation in the kernel computation:

$$
k_{w_{l_1,i_1,j_1}, w_{l_2,i_2,j_2}} = \sigma_k^2 \exp\left(\sum_{d=1}^{D_z} \frac{-(z_{l_1,i_1,d} - z_{l_2,i_2,d})^2}{2l_d^2}\right) \exp\left(\sum_{d=1}^{D_z} \frac{-(z_{l_1+1,j_1,d} - z_{l_2+1,j_2,d})^2}{2l_{D_z+d}^2}\right),
$$

that is, the computation of the covariance matrix between all the weights can be computed efficiently as a product of two sub-covariance matrices that are computed using the latent variables.

## 3.2 Contextual variables for modulating function priors

We first note that while the hierarchical latent variables and meta mappings introduce non-trivial coupling between the weights a priori, the weights and latent variables are inherently global. That is, a function drawn from the model, represented by a set of weights, does not take into account the inputs at which the function will be evaluated. Inspired by recent work on using contextual information or dataset summary to drive supervised learning predictions (Edwards & Storkey, 2016; Garnelo et al., 2018), we introduce the input variable into the weight codes $c_{w_{l,i,j}} = [\mathbf{z}_{l+1,i}, \mathbf{z}_{l,j}, x]$. In turn, this yields input-conditional weight models $p(w_{n,l,i,j} | f, \mathbf{z}_{l+1,i}, \mathbf{z}_{l,j}, \mathbf{x}_n)$. We again turn to compositional kernels and introduce a new **input kernel** $K_x$ which we use as follows,

$$
\begin{aligned}
k_{c_w}(c_{w^1}, c_{w^2}) &= k([\mathbf{z}_{l^1+1,i^1}, \mathbf{z}_{l^1,j^1}, x_1], [\mathbf{z}_{l^2+1,i^2}, \mathbf{z}_{l^2,j^2}, x_2]) \\
&= k_{out}(\mathbf{z}_{l^1+1,i^1}, \mathbf{z}_{l^2+1,i^2}) \cdot k_{in}(\mathbf{z}_{l^1,j^1}, \mathbf{z}_{l^2,j^2}) \cdot k_x(x_1, x_2).
\end{aligned}
\tag{6}
$$

As a result of having private contextual inputs to the meta mapping, the weights are now also local to each data point. In effect, each function drawn from this model explicitly needs the input locations we are interested in, in a similar vein to how functions are drawn from a Gaussian process.

*What effects should we expect from such a modulation?* As usual, the modeler has to choose the input kernel. Consider the use of an exponentiated quadratic kernel: we would expect data which lies far away from training data to receive small kernel values from $K_x$. This, in turn, would modulate the entire kernel $K_{c_w}$ for that data point to small values, leading to a weight model that reverts increasingly to the prior. We would hope such a model would help with modeling uncertainty by resetting weights to uninformative distributions away from training data. However, one may also want to use this mechanism to model inductive biases for the network, such as adding structure to the weight prior that can be captured with a kernel function. This is a potentially appealing avenue, as multiple useful kernels have been found in the GP literature that allow modelers to describe relationships between data, but was not a tool that was previously accessible to neural network modelers. We consider this a novel form of functional regularization, as the entire network can be given structure that will constrain its function space. Unfortunately, we cannot easily apply this method directly to high dimensional inputs as commonly used kernel functions fail in this setting. To overcome this limitation, we introduce $\{\boldsymbol{\epsilon}_n\}_{n=1}^N$, one for each training instance, which can be then used to as an extra input dimension for the meta mapping to generate weights instead of the actual inputs. In detail, each auxiliary input is obtained via a (potentially nonlinear) transformation applied to an input: $\boldsymbol{\epsilon}_n = g(\mathbf{V}\mathbf{x}_n)$, where $\mathbf{V} \in \mathcal{R}^{D_e \times D_x}$, and $D_e$ and $D_x$ are the dimensionality of $\boldsymbol{\epsilon}_n$ and $\mathbf{x}_n$, respectively, and $g(\cdot)$ is an arbitrary transformation. We may also layer these transformations in general. We typically set $D_e \ll D_x$ so this transformation could be thought of as a dimensionality reduction operation. For low dimensional inputs, we can directly set $\boldsymbol{\epsilon}_n = \mathbf{x}_n$. The auxiliary variable is then augmented to the input of the meta mapping as follows,

$$p(w_{n,l,i,j}|f, \mathbf{z}_{l+1,i}, \mathbf{z}_{l,j}, \mathbf{x}_n, \mathbf{V}) = \mathcal{N}(w_{n,l,i,j}; f([\mathbf{z}_{l+1,i}, \mathbf{z}_{l,j}, \boldsymbol{\epsilon}_n]), \sigma_w^2), \qquad (7)$$

that is, the input dimension of the meta mapping is now $2D_z + D_e$. The covariance matrices can be decomposed and computed efficiently as Kronecker products $K_{c_w} = K_{out} \otimes K_{in} \otimes K_x$, as $K_x$ maps over data ($N \times N$) instead of over weights ($W \times W$) like the previous kernels and so their product requires efficient approximation to avoid calculating an object of size ($WN \times WN$).[1] Additionally, we also place a prior over the linear transformation: $p(\mathbf{V}) = \mathcal{N}(\mathbf{V}; \mathbf{0}, \mathbf{I})$.

## 4 INFERENCE AND LEARNING USING STOCHASTIC STRUCTURED VARIATIONAL INFERENCE

Performing inference is challenging due to the non-linearity of the neural network and the need to infer an entire latent function $f$. To address these problems, we derive a structured variational inference scheme that makes use of innovations from inducing point GP approximation literature (Titsias, 2009; Hensman et al., 2013; Quiñonero-Candela & Rasmussen, 2005; Matthews et al., 2016; Bui et al., 2017) and previous work on inferring meta-representations (Karaletsos et al., 2018). As a reminder, we write down the joint density of all variables in the model specified in section 3:

$$p(\mathbf{y}, \mathbf{w}, \mathbf{z}, f, \mathbf{V}|\mathbf{x}) = p(\mathbf{z})p(\mathbf{V})p(f)p(\mathbf{w}|f, \mathbf{z}, \mathbf{V}, \mathbf{x})p(\mathbf{y}|\mathbf{w}, \mathbf{x})$$

$$= p(\mathbf{z})p(\mathbf{V})p(f)\left[\prod_{n=1}^N p(\mathbf{w}_n|f, \mathbf{C}_w(\mathbf{z}, \mathbf{x}_n, \mathbf{V}))p(\mathbf{y}_n|\mathbf{w}, \mathbf{x}_n)\right].$$

We first partition the space $\mathcal{Z}$ of inputs to the function $f$ into a finite set of $M$ variables called inducing inputs $\mathbf{z}_\mathbf{u}$ and the remaining inputs, $\mathcal{Z} = \{\mathbf{z}_\mathbf{u}, \mathcal{Z}_{\neq \mathbf{z}_\mathbf{u}}\}$. The function $f$ is partitioned identically, $f = \{\mathbf{u}, f_{\neq \mathbf{u}}\}$, where $\mathbf{u} = f(\mathbf{z}_\mathbf{u})$. We can then rewrite the GP prior as follows, $p(f) = p(f_{\neq \mathbf{u}}|\mathbf{u}, \mathbf{z}_\mathbf{u})p(\mathbf{u}|\mathbf{z}_\mathbf{u})$.[2] The inducing inputs and outputs, $\{\mathbf{z}_\mathbf{u}, \mathbf{u}\}$, will be used to parameterize the approximation. In particular, a variational approximation is judiciously chosen to mirror the form of the joint density:

$$q(\mathbf{w}, \mathbf{z}, f, \mathbf{V}|\mathbf{x}) = q(\mathbf{z})q(\mathbf{V})p(f_{\neq \mathbf{u}}|\mathbf{u}, \mathbf{z}_\mathbf{u})q(\mathbf{u})p(\mathbf{w}|f, \mathbf{z}, \mathbf{V}, \mathbf{x}), \qquad (8)$$

where the variational distribution over $\mathbf{w}$ is made to explicitly depend on remaining variables through the conditional prior, $q(\mathbf{z})$ and $q(\mathbf{V})$ are chosen to be diagonal (mean-field) Gaussian densities, $q(\mathbf{z}) = \mathcal{N}(\mathbf{z}; \boldsymbol{\mu}_\mathbf{z}, \mathrm{diag}(\boldsymbol{\sigma}_\mathbf{z}^2))$ and $q(\mathbf{V}) = \mathcal{N}(\mathbf{V}; \boldsymbol{\mu}_\mathbf{V}, \mathrm{diag}(\boldsymbol{\sigma}_\mathbf{V}^2))$, respectively, and $q(\mathbf{u})$ is chosen to be a correlated multivariate Gaussian, $q(\mathbf{u}) = \mathcal{N}(\mathbf{u}; \boldsymbol{\mu}_\mathbf{u}, \Sigma_\mathbf{u})$. This approximation allows convenient

---

[1]We can solve this in complexity dominated by the decomposition of the largest kernel in the product.

[2]The conditioning on $\mathcal{Z}_{\neq \mathbf{z}_\mathbf{u}}$ in $p(f_{\neq \mathbf{u}}|\mathbf{u}, \mathbf{z}_\mathbf{u})$ is made implicit here and in the rest of this paper.

cancellations yielding a tractable variational lower bound as follows,

$$
\begin{aligned}
\mathcal{L}_{\text{metaGP}}(\cdot) &= \int_{\mathbf{w},\mathbf{z},f,\mathbf{V}} q(\mathbf{w},\mathbf{z},f,\mathbf{V}|\mathbf{x}) \log \frac{p(\mathbf{z})p(\mathbf{V})\cancel{p(f_{\neq\mathbf{u}}|\mathbf{u},\mathbf{z_u})}p(\mathbf{u}|\mathbf{z_u})\cancel{p(\mathbf{w}|f,\mathbf{z},\mathbf{V},\mathbf{x})}p(\mathbf{y}|\mathbf{w},\mathbf{x})}{q(\mathbf{z})q(\mathbf{V})\cancel{p(f_{\neq\mathbf{u}}|\mathbf{u},\mathbf{z_u})}q(\mathbf{u})\cancel{p(\mathbf{w}|f,\mathbf{z},\mathbf{V},\mathbf{x})}} \\
&\approx -\text{KL}[q(\mathbf{z})||p(\mathbf{z})] - \text{KL}[q(\mathbf{V})||p(\mathbf{V})] - \text{KL}[q(\mathbf{u})||p(\mathbf{u}|\mathbf{z_u})] \\
&\quad + \frac{1}{K}\sum_{k=1}^{K}\int_{\mathbf{w},f} q(\mathbf{w},f|\mathbf{z}_k,\mathbf{V}_k)\log p(\mathbf{y}|\mathbf{w},\mathbf{x})
\end{aligned}
$$

where the last expectation has been partly approximated using simple Monte Carlo with the reparameterization trick, i.e. $\mathbf{z}_k \sim q(\mathbf{z})$ and $\mathbf{V}_k \sim q(\mathbf{V})$. We will next discuss how to approximate the expectation $\mathcal{F}_k = \int_{\mathbf{w},f} q(\mathbf{w},f|\mathbf{z}_k,\mathbf{V}_k,\mathbf{x})\log p(\mathbf{y}|\mathbf{w},\mathbf{x})$. Note that we split f into $f_{\neq\mathbf{u}}$ and $\mathbf{u}$, and that we can integrate $f_{\neq\mathbf{u}}$ out exactly to give, $q(\mathbf{w}|\mathbf{z}_k,\mathbf{u},\mathbf{V}_k,\mathbf{x}) = \mathcal{N}(\mathbf{w};\mathbf{A}^{(k)}\mathbf{u},\mathbf{B}^{(k)})$,

$$
\mathbf{A}^{(k)} = \mathbf{K}^{(k)}_{f_{\neq\mathbf{u}}\mathbf{u}}\mathbf{K}^{-1}_{\mathbf{uu}}; \qquad \mathbf{B}^{(k)} = \mathbf{K}^{(k)}_{l,f_{\neq\mathbf{u}}f_{\neq\mathbf{u}}} - \mathbf{K}^{(k)}_{f_{\neq\mathbf{u}}\mathbf{u}}\mathbf{K}^{-1}_{\mathbf{uu}}\mathbf{K}^{(k)}_{\mathbf{u}f_{\neq\mathbf{u}}} + \sigma_w^2\mathbf{I}. \tag{9}
$$

At this point, we can either (i) sample $\mathbf{u}$ from $q(\mathbf{u})$, or (ii) integrate $\mathbf{u}$ out analytically. We opt for the second approach, which gives

$$
q(\mathbf{w}|\mathbf{z}_k,\mathbf{V}_k,\mathbf{x}) = \mathcal{N}(\mathbf{w};\mathbf{A}^{(k)}\boldsymbol{\mu}_u,\mathbf{B}^{(k)} + \mathbf{A}^{(k)}\Sigma_{\mathbf{u}}\mathbf{A}^{\mathsf{T},(k)}). \tag{10}
$$

In contrast to GP regression and classification in which the likelihood term is factorized point-wise w.r.t. the parameters and thus their expectations only involve a low dimensional integral, we have to integrate out $\mathbf{w}$ in this case, which is of much higher dimensions. When necessary or practical, we resort to Kronecker factored models or make an additional diagonal approximation as follows,

$$
\hat{q}(\mathbf{w}|\mathbf{z}_k,\mathbf{V}_k,\mathbf{x}) = \mathcal{N}(\mathbf{w};\mathbf{A}^{(k)}\boldsymbol{\mu}_u,\text{diag}(\mathbf{B}^{(k)} + \mathbf{A}^{(k)}\Sigma_{\mathbf{u}}\mathbf{A}^{\mathsf{T},(k)})). \tag{11}
$$

Whilst the diagonal approximation above might look poor from the first glance, it is conditioned on a sample of the latent variables $\mathbf{z}_k$ and thus the weights' correlations are retained after integrating out $\mathbf{z}$. Such correlation is illustrated in fig. 2 where we show the marginal and conditional covariance structures for the weights of a small neural network, separated into diagonal and full covariance models. The diagonal approximation above has been observed to give pathological behaviours in the GP regression case (Bauer et al., 2016), but we did not observe these in practice. A comparison of this diagonal approximation to alternatives is provided in the appendix. $\mathcal{F}_k$ is approximated by $\mathcal{F}_k \approx \int_{\mathbf{w}} \hat{q}(\mathbf{w}|\mathbf{z}_k,\mathbf{V}_k,\mathbf{x})\log p(\mathbf{y}|\mathbf{w},\mathbf{x})$ which can be subsequently efficiently estimated using the *local reparameterization trick* (Kingma et al., 2015). The final lower bound is then optimized to obtain the variational parameters of $q(\mathbf{u})$, $q(\mathbf{z})$ and $q(\mathbf{V})$, and estimates for the noise in the meta-GP model, the kernel hyper-parameters and the inducing inputs.

## 5 RELATED WORK

Bayesian neural networks have recently garnered a resurgence of interests in the community. There is, however, a long history of research on developing approximate Bayesian inference methods for these networks. Notable work include extended Kalman filtering (Singhal & Wu, 1989), Laplace's approximation (MacKay, 1992), Hamiltonian Monte Carlo (Neal, 1993; 2012), variational inference (Hinton & Van Camp, 1993; Barber & Bishop, 1998; Graves, 2011; Blundell et al., 2015; Gal & Ghahramani, 2016), and approximate expectation propagation (Hernández-Lobato & Adams, 2015; Li et al., 2015; Hernández-Lobato et al., 2016). The work in this paper is orthogonal to these as the model employs a hierarchical prior, and inference is done in a lower-dimensional latent space instead of the weight space. The variational approximation is chosen such that the marginal distribution over the weights is non-Gaussian and the correlations between weights are retained, in contrast to the popular mean-field Gaussian approximation. Additionally, a standard multivariate Normal distribution is typically used as the prior over the network weights and biases. Imposing additional structures over the weights a prior or carefully choosing the right prior variances have been observed to improve the predictive performance (Ghosh et al., 2018; Neal, 2012; Blundell et al., 2015). In this vein, the proposed model in this paper imposes a class of flexible structures over the network parameters and learn a posterior over them conditioned on observed data. Another related theme is hyper-networks, the core idea of which is to generate network parameters using another network (see e.g. Ha et al., 2016; Stanley et al., 2009). The proposed model could be thought of as a GP

hyper-network, but utilizes latent variables for nodes as introduced in (Karaletsos et al., 2018), thus avoiding over-parametrized, monolithic models over weight tensors. Instead, the model uses hierarchical modeling, conditional probabilities, and the unit nodes to perform one-dimensional regression to weights maintaining modeling correlations.

# 6 EXPERIMENTS

In this section, we evaluate the proposed model and inference scheme on several toy and benchmark datasets. We compare the proposed approach primarily to maximum a posteriori estimation (MAP) and MFVI, both of which use a standard Normal prior over the weights. These were implemented using PyTorch (Paszke et al., 2017) and the code will be available upon acceptance. Additional results are included in the appendices. We use $M = 50$ inducing points for all experiments in this section.

## 6.1 TOY REGRESSION AND CLASSIFICATION EXAMPLES

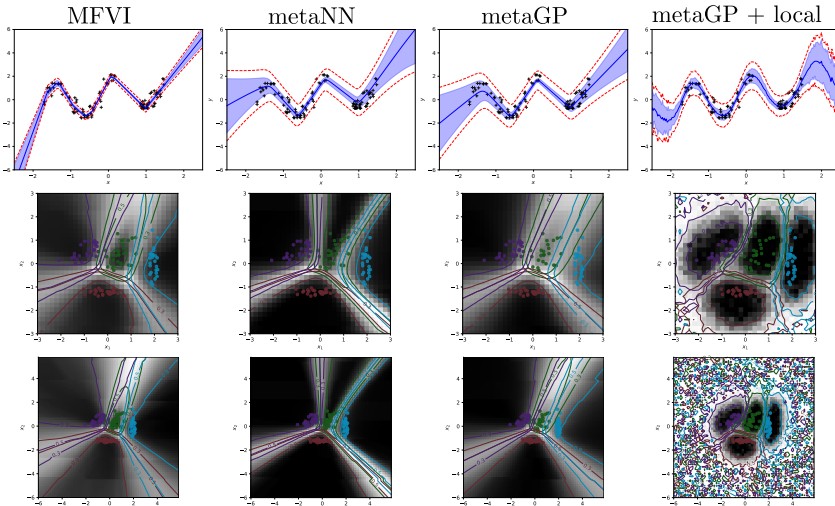

Figure 3: Illustration of the performance of Bayesian neural networks with a standard Normal prior over the weights and a mean-field Gaussian variational approximation [MFVI], a meta-NN prior, a meta-GP prior and a meta-GP prior with auxiliary inputs. Top: 1-D regression on 100 noisy observations. The plots show the predictive mean and uncertainty as well as the training points. Middle and bottom: Multiclass classification on a toy four-class dataset of 100 examples. The background colour matches the entropy of the predictive probability distribution, e.g. darker is more certain. Each pair of plots show the behaviour around and away from the training points when zooming out. Best viewed in colour.

We first illustrate the performance of the proposed model on two toy examples, one regression and one classification. In both cases, a Bayesian neural network with one hidden layer of 50 hidden units is used. Additionally, the hierarchical latent variables have two dimensions. For the regression case, we generate 100 data points from a toy function suggested by Louizos et al. (2019). For classification, a four-class dataset of 100 data points is used. Figure 3 shows the predictive performance of the proposed models, MFVI and a meta-NN on both tasks. Notably, the proposed model with contextual inputs (and thus input dependent network weights) gives uncertainty estimates that are reminiscent to that of a GP regression model, despite being a neural network under the hood. Following Bradshaw et al. (2017), we also show the uncertainty for data further from the training instances. The model with auxiliary inputs remains uncertain as expected for these points whilst MFVI produces arguably overconfident predictions. We also attempt to summarize the distribution of the weights in the network in appendix A.7 in the appendices.

## 6.2 INPUT DEPENDENT NEURAL NETWORKS FOR UNCERTAINTY QUANTIFICATION

Motivated by the performance of the proposed metaGP model with auxiliary inputs in the toy examples in fig. 3, this section tests the ability of this model class to produce calibrated predictive uncertainty to out-of-distribution samples. That is, for test samples that do not come from the

same training distribution, a robust and well-calibrated model should produce uncertain predictive distribution and thus high predictive entropy. Such a model could find applications in safety-critical tasks where the cost of making confident but wrong predictions is high, or in area where detecting unfamiliar inputs is crucial such as active learning or reinforecement learning. In this work, we first train a neural network classifier with one hidden layer of 100 rectified linear units on the MNIST dataset, and apply the metaGP prior only to the last layer of the network. The dimensions of the latent variables and the auxiliary inputs are both 2. After training, we compute the entropy of the predictions on various test sets, including notMNIST, fashionMNIST, Kuzushiji-MNIST, and uniform and Gaussian noise inputs. Following (Lakshminarayanan et al., 2017; Louizos & Welling, 2017), the CDFs of the predictive entropies for various methods are shown in fig. 4. A calibrated classifier should give a CDF that bends towards the top-left corner of the plot for in-distribution examples and, vice versa, towards the bottom-right corner of the plot for out-of-distribution inputs. In most out-of-distribution sets considered, except Gaussian random noise, metaGP and metaGP with local auxiliary variables demonstrate competitive performance to Gaussian MFVI. Notably, MAP estimation, often deployed in practice, tends to give wildly poor uncertainty estimates on out-of-distribution samples. We illustrate this behaviour and that of other models and methods on representative inputs of the MNIST and Kuzushiji-MNIST datasets in fig. 4.

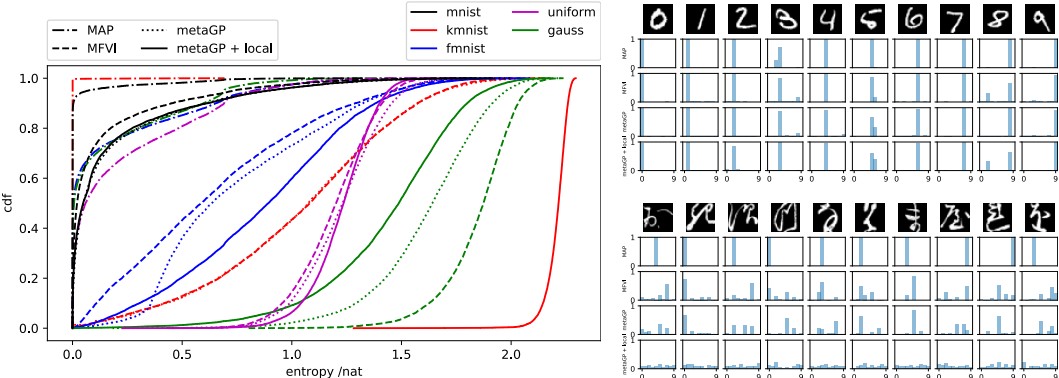

Figure 4: The CDFs of predictive entropies on in-distribution and out-of-distribution test sets for various methods [Left] and the predictive class probability by these methods for representative samples from an in-distribution test set [Top right] and an out-of-distribution test test [Bottom right]. Best viewed in colour.

### 6.3 Modeling Inductive Biases For Neural Networks With Input-Dependent Kernels

We further explore the utility of the contextual variable towards modeling inductive biases for neural networks and evaluate on predictive performance on a regression example. In particular, we generate 100 training points from a synthetic sinusoidal function and create two test sets that contains in-sample inputs and out-of-sample inputs, respectively. We test an array of models and inference methods, including BNN with MFVI, metaGP and metaGP with contextual variables. In particular, as noted in section 3.1 we can choose the covariance function to be used for the auxiliary variables to encode our belief about how the weights should be modulated by the input. We pick exponentiated quadratic and periodic kernels (MacKay, 1998) in this example. The predictive performance are measured by the root mean squared error (RMSE) and the log-likelihood (LL) on the test examples. Figure 5 summarizes the results and illustrate the qualitative difference between models. It could be noted that the periodic kernel allows the model to discover and encode periodicity, allowing for more long-range confident predictions compared to that of the exponentiated quadratic kernel. In this example, MetaGP with contextual variables is superior to other methods, demonstrating good RMSE and LL on both in-distribution and out-of-distribution examples.

### 6.4 Active Learning

We next stress-test the performance of the proposed model in a pool-based active learning setting for real-valued regression, where limited training data is provided initially and the target is to sequentially select points from a pool set to add to the training set. The criterion to select the next best point from

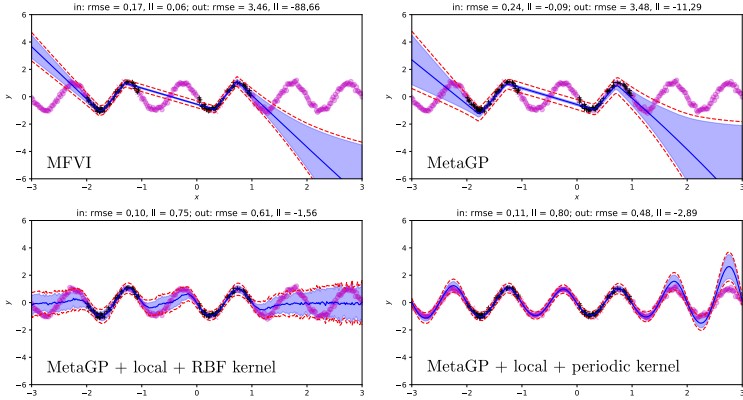

Figure 5: Illustration of the effect of local variables and different kernels for the local variables. The underlying function is a sinusoid. Each subplot shows the training points (black pluses), the in-distribution test points (magenta crosses), the out-of-distribution test points (magenta filled circles), the predictive mean (connected line), two standard deviations (shaded area) and added observation noise (red dashed line). The title of the subplots include the predictive performance on the in-distribution and out-of-distribution test points, demonstrating the ability of a model to interpolate and extrapolate, correspondingly. Best viewed in colour.

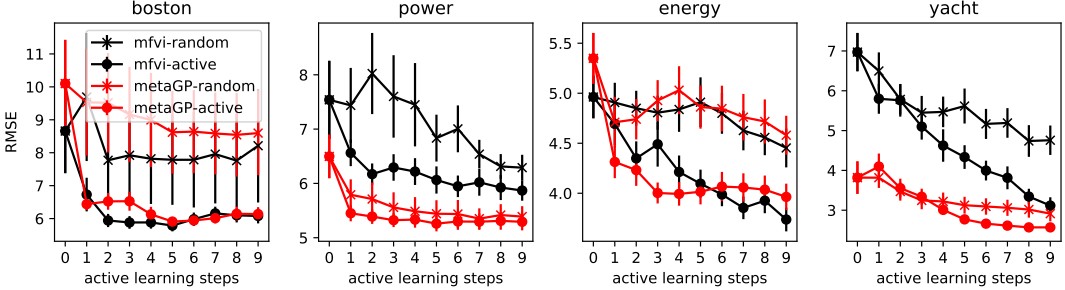

Figure 6: Active learning with BNNs using mean-field Gaussian variational inference [MFVI] and a meta-GP hierarchical prior [MetaGP] on several UCI regression datasets. Each trace shows the root mean squared error (RMSE) on the test set as more data points are selected and moved from the pool set to the training set, averaged over 40 runs. The objective function for selecting points from the pool set is the predictive variance. Best viewed in colour.

the pool set is based on the entropy of the predictive distribution, i.e. we pick one with the highest entropy. In practice, we approximate the predictive density by a Gaussian density, which results in a tractable entropy computation. Note that this selection procedure can be interpreted as selecting points that maximally reduce the posterior entropy of the network parameters (Houlsby et al., 2011). Four UCI regression datasets were considered, where each with 40 random train/test/pool splits. For each split, the initial train set has 20 data points, the test set has 100 data points, and the remaining points are used for the pool set, similar to the active learning set-up in Hernández-Lobato & Adams (2015). We compare the performance of the proposed model and inference scheme to that of Gaussian mean-field variational inference and show the average results in fig. 6. Across all runs, we observe that active learning is superior to random selection and more crucially using the proposed model and inference scheme seems to yield comparable or better predictive errors with a similar number of queries. We also illustrate the procedure on a toy classification dataset and visualize the predictive uncertainty as more points are picked and used for training in fig. 17 in the appendix. This simple setting quantitatively reveals the inferior performance of MFVI, compared to MAP and metaGP.

## 7 SUMMARY

We have developed a flexible hierarchical neural network model and a structured variational inference scheme that together (i) enable flexible modelling of the correlations between network parameters, (ii) allow weight structures to be retained in the approximate posterior without having to perform direct inference in the high-dimensional weight space, and (iii) allow input-dependent contextual variables to be used to generate private weights per data point which translates to more desirable

uncertainty estimates in practice and the ability to generalize out of sample with inductive biases such as periodicity. We plan to evaluate the performance of the model on more challenging decision making tasks such as contextual bandits or reinforcement learning, and to extend the inference scheme to handle continual learning.

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

# A   EXTRA EXPERIMENTAL RESULTS

## A.1   AN EMPIRICAL EVALUATION OF VARIOUS APPROXIMATIONS FOR $q(\mathbf{w}|\mathbf{z}_k, \mathbf{V}_k, \mathbf{x})$

In this experiment, we analyze the impact of different approximations to the covariance matrix of $q(\mathbf{w}|\mathbf{z}_k, \mathbf{V}_k, \mathbf{x})$:

$$q(\mathbf{w}|\mathbf{z}_k, \mathbf{V}_k, \mathbf{x}) = \mathcal{N}(\mathbf{w}; \mathbf{A}^{(k)}\boldsymbol{\mu}_u, \mathbf{B}^{(k)} + \mathbf{A}^{(k)}\Sigma_{\mathbf{u}}\mathbf{A}^{\intercal,(k)}).$$

If we use the exact, fully correlated Gaussian distribution above, it is necessary to sample from this distribution to evaluate the lower bound. This step costs $\mathcal{O}(W^3)$ where $W$ is the number of parameters in the network.

The complexity can be greatly improved by making a diagonal approximation to $\mathbf{B}^{(k)}$ as follows,

$$\hat{q}_{\text{FITC}}(\mathbf{w}|\mathbf{z}_k, \mathbf{V}_k, \mathbf{x}) = \mathcal{N}(\mathbf{w}; \mathbf{A}^{(k)}\boldsymbol{\mu}_u, \text{diag}(\mathbf{B}^{(k)}) + \mathbf{A}^{(k)}\Sigma_{\mathbf{u}}\mathbf{A}^{\intercal,(k)}).$$

Sampling from this distribution can be done in $\mathcal{O}(WM^2)$ where M is the number of pseudo-points.

This can be further approximated by assuming a diagonal covariance matrix,

$$\hat{q}_{\text{diag}}(\mathbf{w}|\mathbf{z}_k, \mathbf{V}_k, \mathbf{x}) = \mathcal{N}(\mathbf{w}; \mathbf{A}^{(k)}\boldsymbol{\mu}_u, \text{diag}(\mathbf{B}^{(k)} + \mathbf{A}^{(k)}\Sigma_{\mathbf{u}}\mathbf{A}^{\intercal,(k)})).$$

The variational bound can then be evaluated by drawing samples from $\hat{q}_{\text{diag}}$ as in the above approximation, or by drawing activity samples by employing the local reparameterization trick (Kingma et al., 2015).

We evaluate the performance of using the exact and approximate conditional distributions above in a range of toy regression and classification, and show representative results in figs. 7 and 8. We note that the diagonal approximation is fast and gives qualitatively similar performance compared to more structured approximation or the exact case, in both cases where there is a singla GP for all weights in the network and there is multiple GPs, one for each weight layer in the network.

## A.2   ROBUSTNESS IN VARIOUS DATA REGIMES FOR A TOY REGRESSION PROBLEM

In this experiment, we evaluate the qualitative performance of various methods, including MFVI, NN-Metarep, GP-Metarep and GP-Metarep with local, input-dependent kernel, on a toy regression problem, in different data regimes. In particular, we considers 10, 20, and 50 training points respectively, and plot the predictions in fig. 9. GP-metarep demonstrates consistent performance across all data regimes, comparable to that of NN-metarep. The input-dependent kernel helps the performance further in the out-of-distribution area.

## A.3   EFFECT OF INPUT-DEPENDENT KERNEL AND APPLICATIONS TO FORWARD TRANSFER/FAST ADAPTATION/MULTI-TASK LEARNING

In this section, we investigate the effect of the input-dependent kernel by probing the learnt kernel hyperparameters. In particular, we first train a model with an periodic input-dependent kernel on a sinusoidal 1D data set, as show in the main text. We then vary the period hyper-parameter in the kernel whilst keeping other hyperparameters and variational parameters fixed. The predictions for a few hyperparameters are shown in fig. 10. We note the variation/period in the data is captured by weight modulation, governed by the input-dependent kernel. Changing the period hyperparameter affects how fast or slow the weights are changing wrt the input.

We further investigate using the proposed model for multi-task learning. In particular, the latent variable $\mathbf{z}$ and corresponding hyper-parameters and variational parameters can be shared across different tasks whilst the meta mapping and the input-dependent kernel are private to each individual task. We first train the model on four regression tasks, each corresponds to a sinusoid of a particular frequency. At test time, a novel test set is shown to the model. The hyper-parameters of the input kernel and variational parameters corresponding to this new test set are optimised while other hyper-parameters and the latent variables are kept fixed. We evaluate the performance of the model on the novel test sets to see how the latent variables can be reused and shared across tasks to facilitate fast adaptation to new settings. The performance of the model on the tasks used for training and new tasks at test time is shown in fig. 11. This result demonstrates the ability of the model trained with multiple similarly related tasks to faithfully and quickly adapt to new settings.

## A.4   MNIST EXPERIMENT: A COMPARISON TO DEEP KERNEL LEARNING AND FULL FIGURES

In this section, we include the full figures of the MNIST out-of-distribution uncertainty experiment, as well as additional results using deep kernel learning (Wilson et al., 2016). In particular, we employ the same network architecture with the last layer being replaced by multiple independent GPs, one for each class (output dimension). As exact inference is intractable, variational inference based on inducing points is employed – we used 50 inducing points for each output. The full results of all models/methods considered are shown in fig. 12. For clarify, the results of deep kernel learning and GP-metarep are shown in fig. 13. GP-metarep with the input-dependent kernel shows good performance, outperforming deep kernel learning in all cases. In addition, we include the full figures for the predictive distributions on representative test examples in figs. 14 and 15.

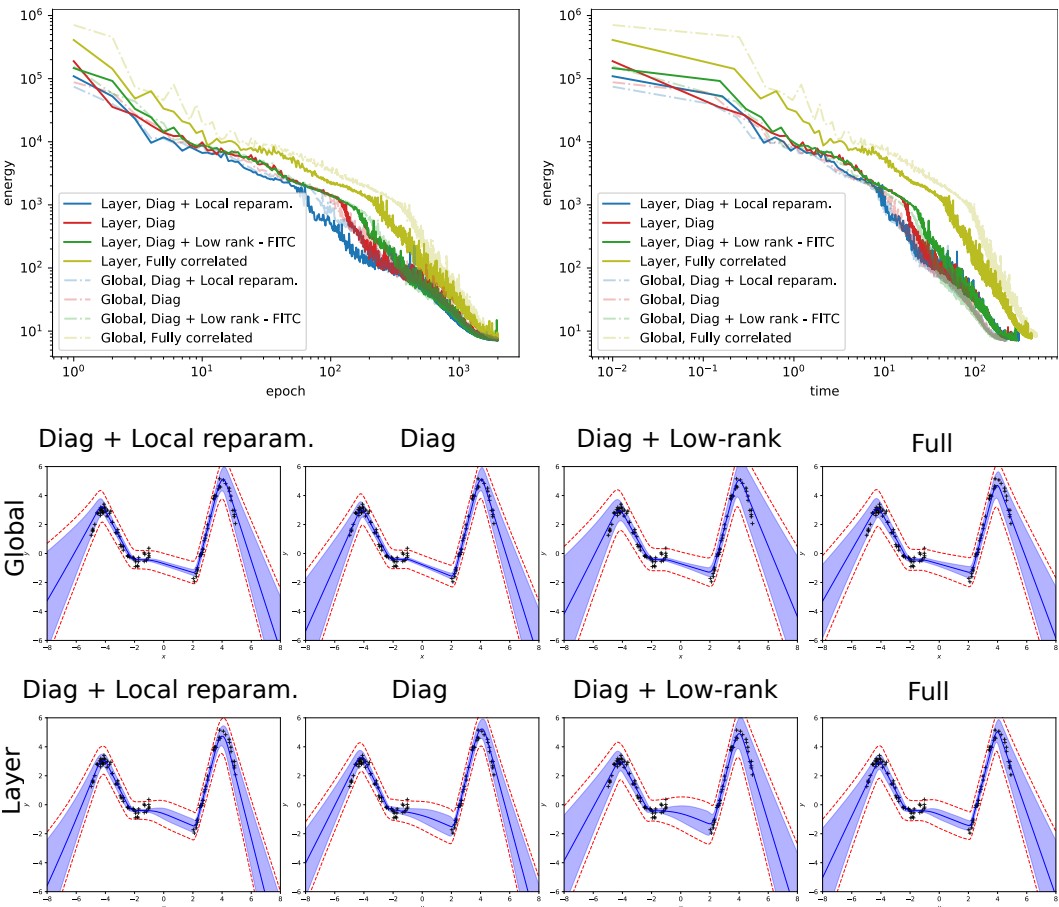

Figure 7: An evaluation of the covariance matrix approximations in a toy regression example. Top: objective function during training vs epoch/time. Bottom: Predictions after training using one of the approximations discussed in the text. Global: there is one GP for all weights in the network. Layer: there are multiple GPs, one for each weight layer in the network. Best viewed in colour.

### A.5 ROBUSTNESS OF GP-METAREP WITH NETWORK ARCHITECTURES

In this experiment, we compare the performance of GP-metarep for various numbers of hidden units (20, 50 and 100) and two activation functions (tanh and ReLU) on a toy regression problem. The observation noise is fixed in this experiment. We observe that the performance of the models is in general consistent across different activation functions and numbers of hidden units. We show the results in fig. 16.

### A.6 A TOY ACTIVE LEARNING SETTING

Please see fig. 17 and the associated caption.

### A.7 PROBABILITY DENSITIES OF WEIGHTS IN THE TOY 1D REGRESSION EXPERIMENT

Please see figs. 18 and 19 and the associated captions.

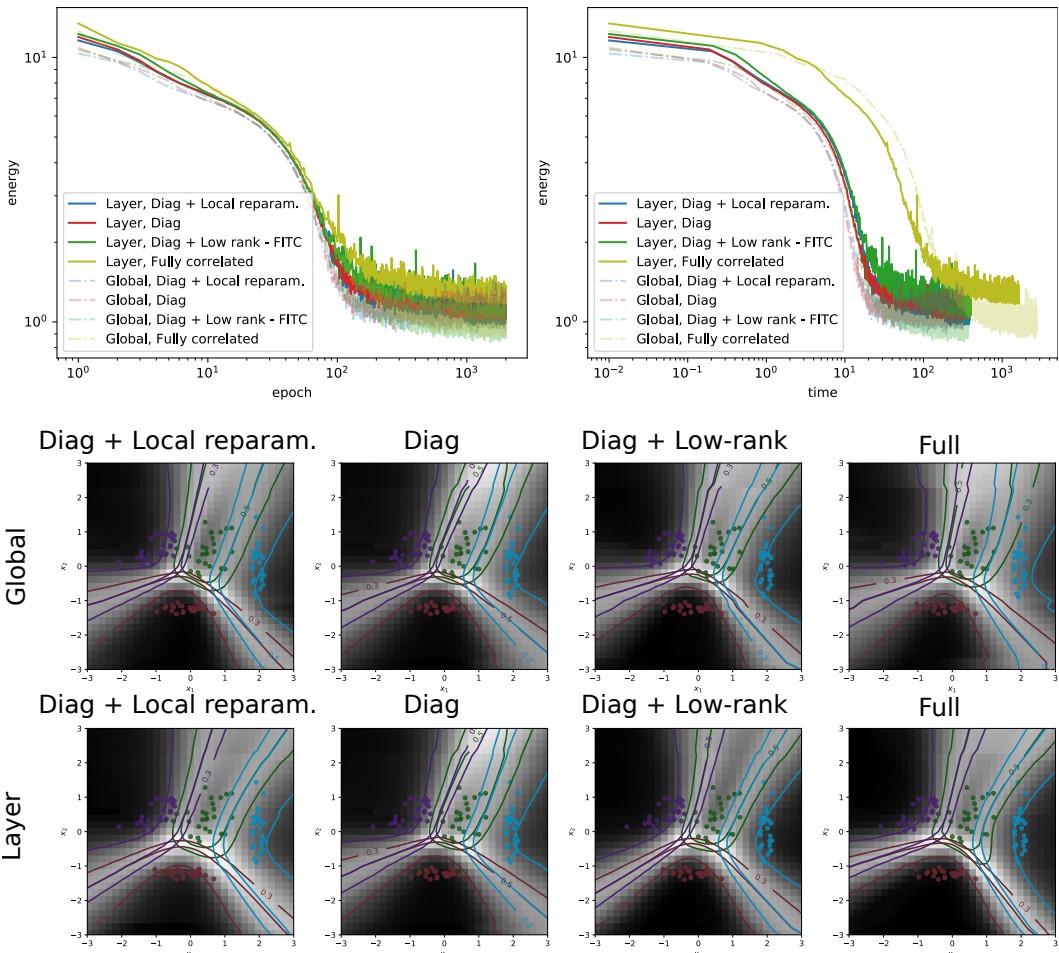

Figure 8: An evaluation of the covariance matrix approximations in a toy classification example. Top: objective function during training vs epoch/time. Bottom: Predictions after training using one of the approximations discussed in the text. Global: there is one GP for all weights in the network. Layer: there are multiple GPs, one for each weight layer in the network. Best viewed in colour.

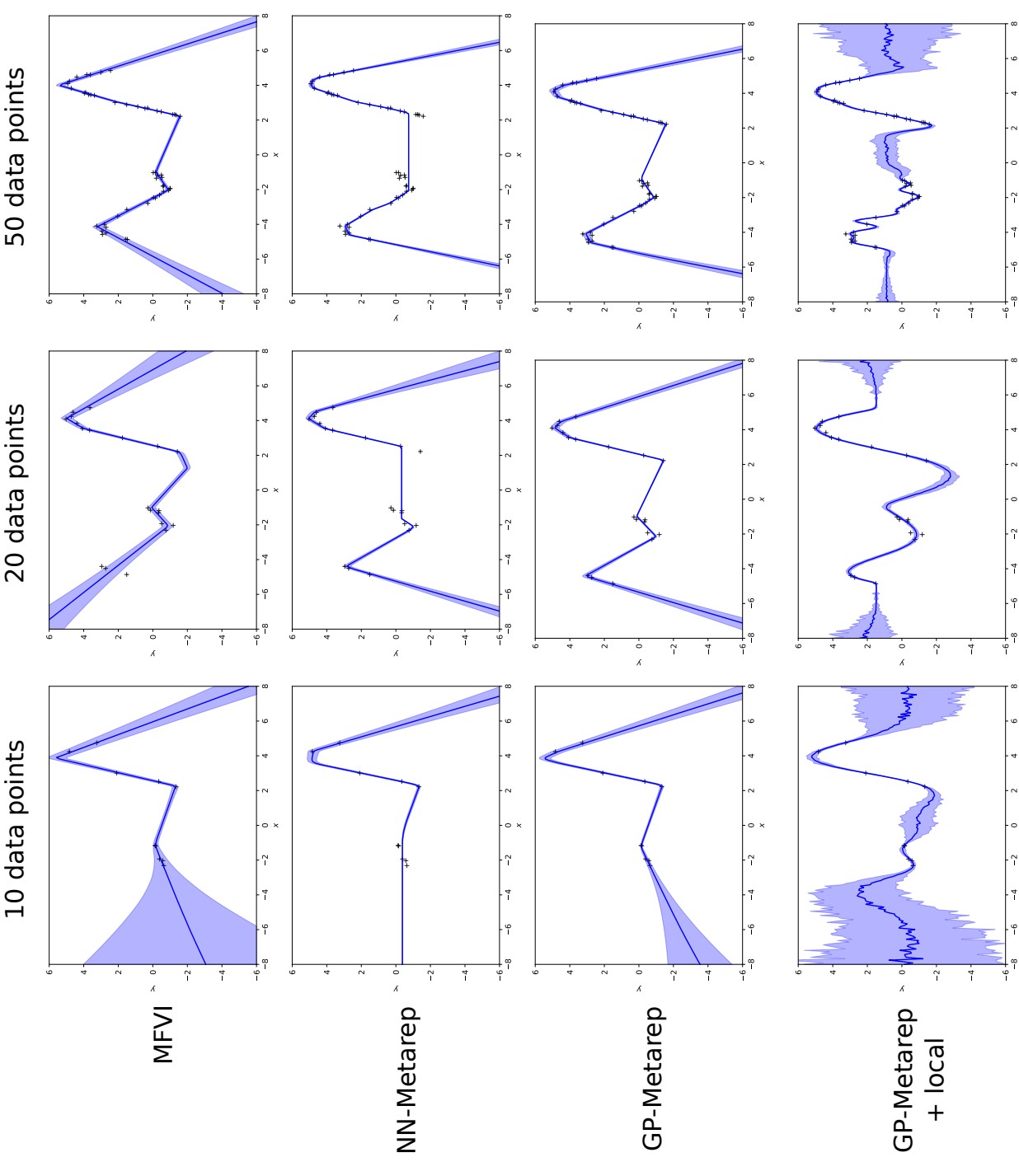

Figure 9: Performance of mean-field variational inference, NN-Metarep with variational inference and GP-Metarep with variational inference on a toy regression problem with various number of training points. Best viewed in colour.

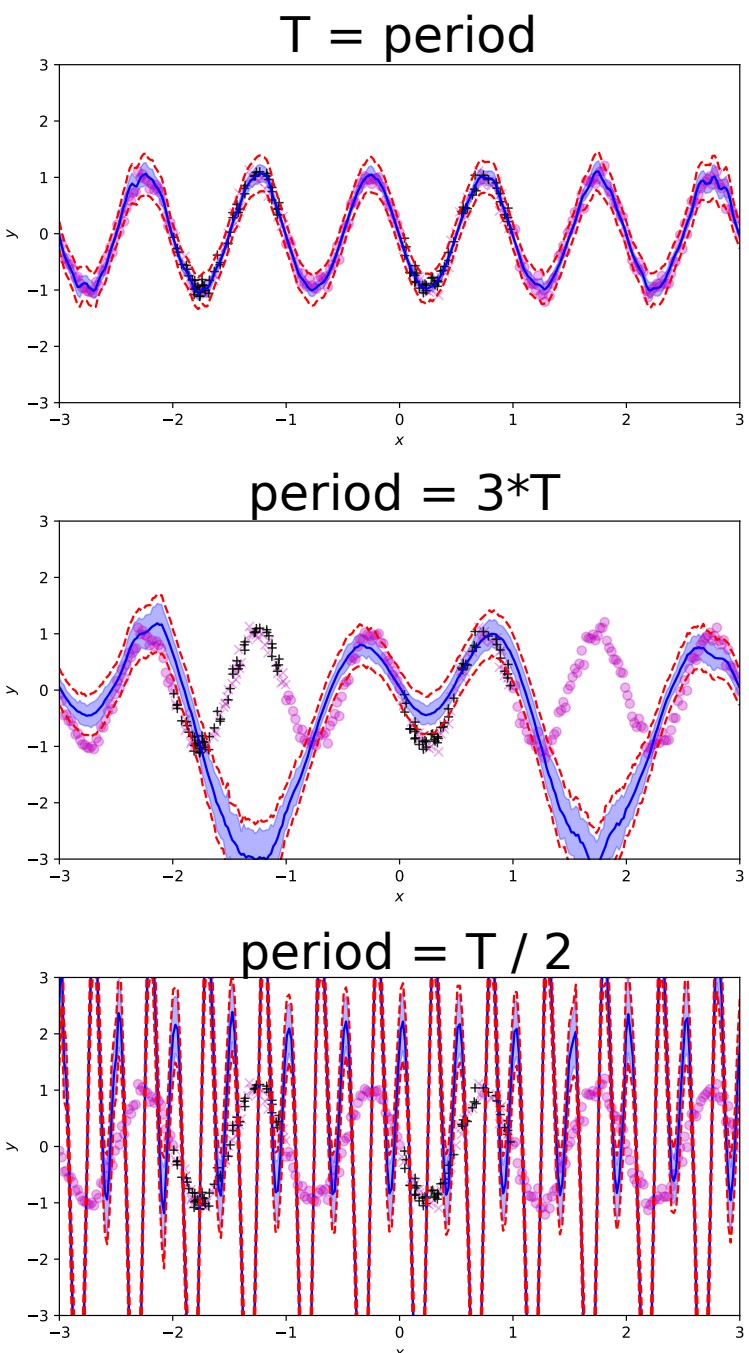

Figure 10: We first train a model with an input-dependent kernel on a sinusoid data set (top) and then vary the period hyperparameter of the input-dependent kernel whilst keeping other hyperparameters and variational parametes fixed (middle and bottom). Best viewed in colour.

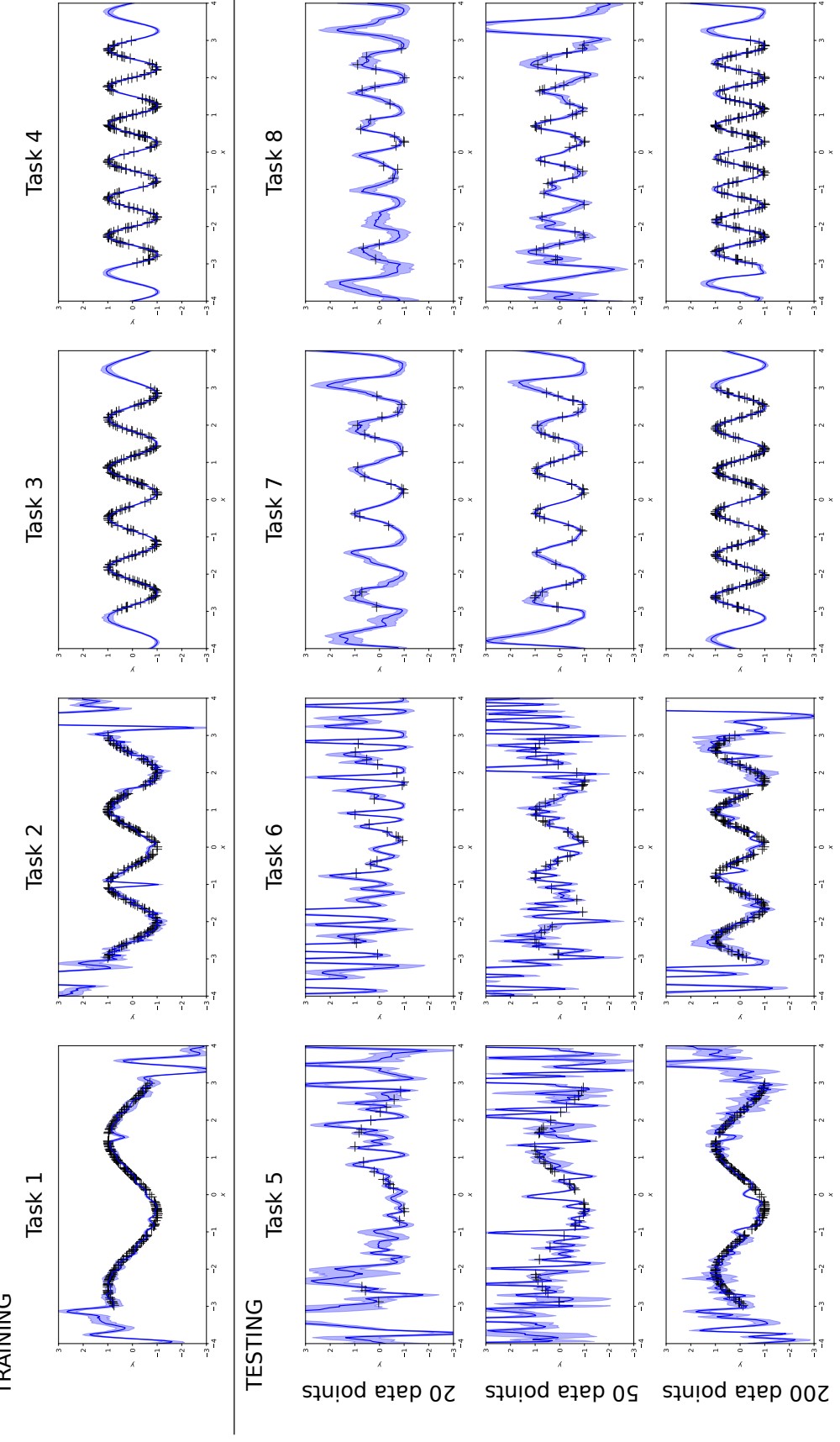

Figure 11: Training on multiple related tasks and adaptation to novel tasks at test time. In this case, the latent variables (as well as weight code hyperparameters) are shared across tasks while each individual has its own input-dependent kernel. At test time, only the private parameters for the new task are re-initialised and optimised. Best viewed in colour.

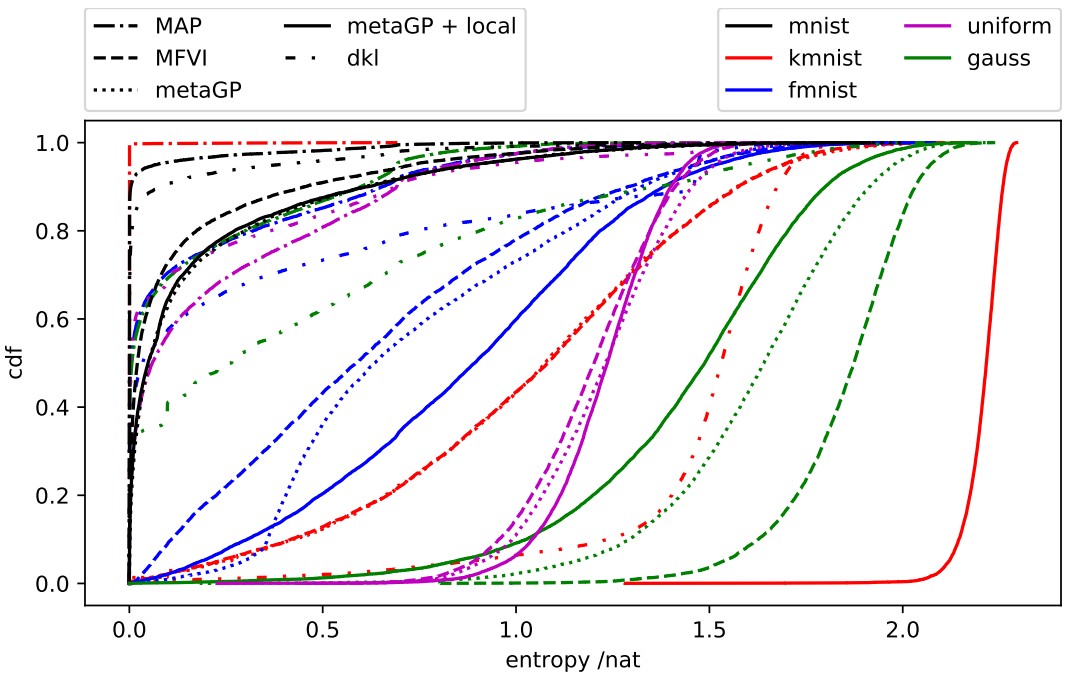

Figure 12: Full results of the MNIST out-of-distribution uncertainty experiment. Best viewed in colour.

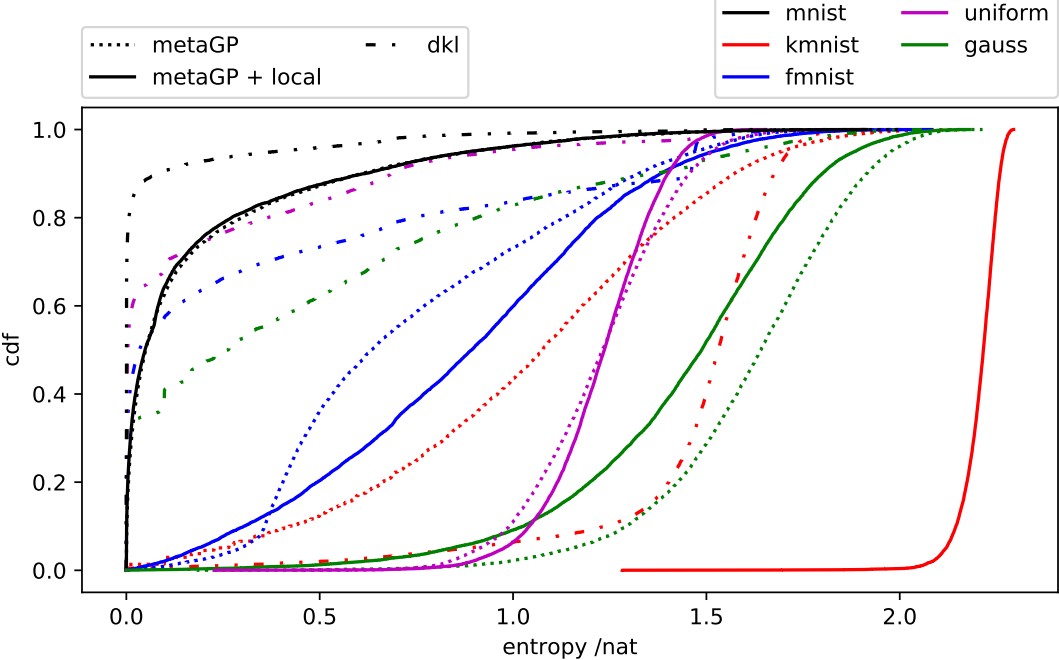

Figure 13: For clarify, we show a subset of the results of the MNIST out-of-distribution uncertainty experiment in fig. 12, for GP-metarep and deep kernel learning. Best viewed in colour.

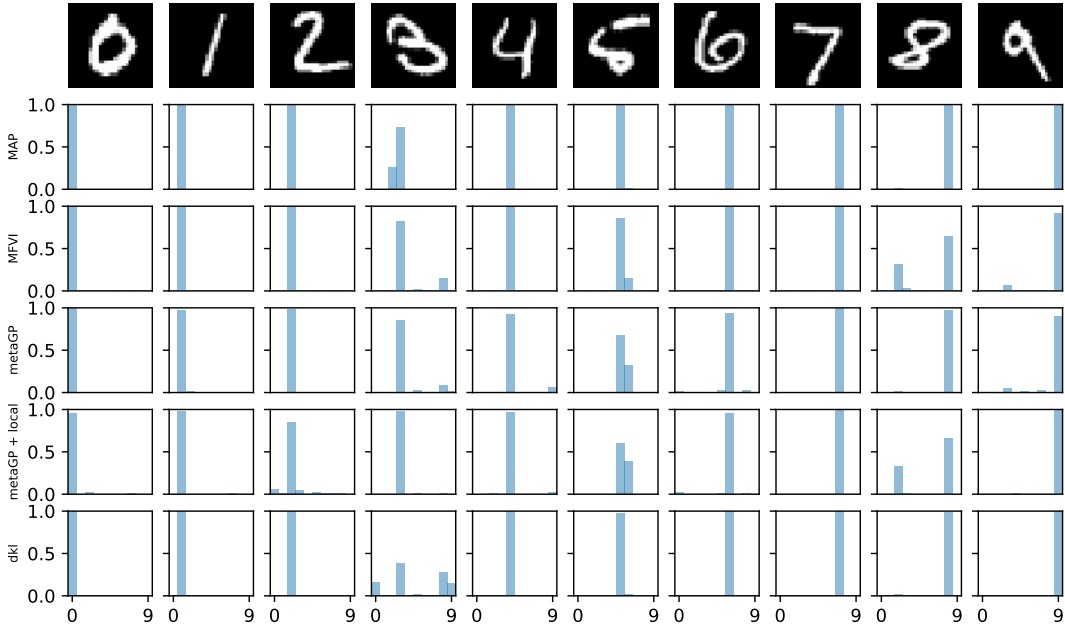

Figure 14: Predictive distribution for representative MNIST test examples by various methods. Best viewed in colour.

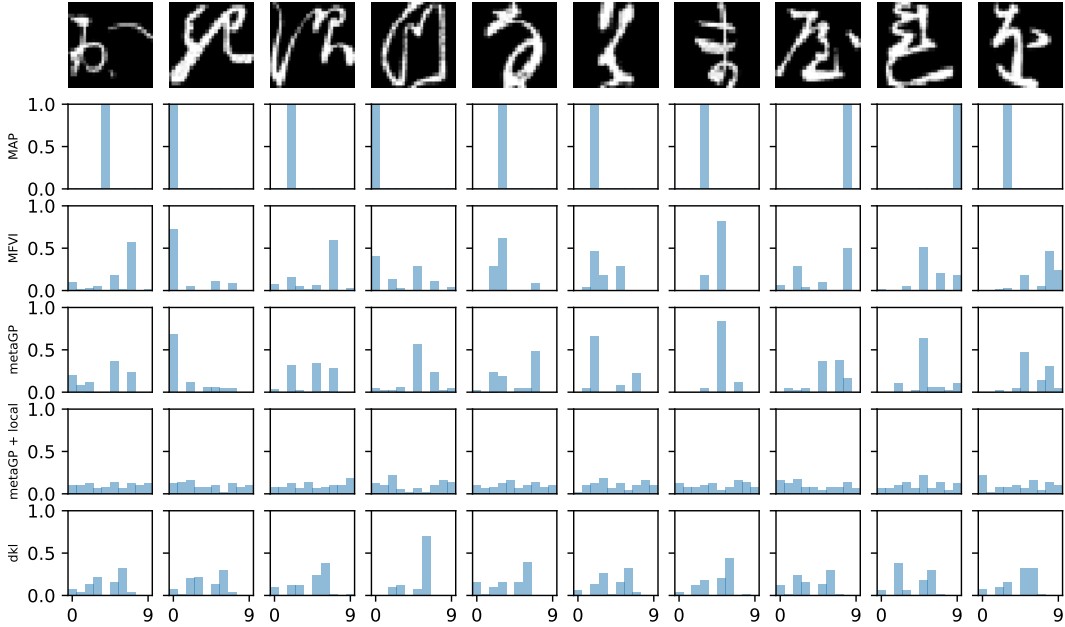

Figure 15: Predictive distribution for representative KMNIST test examples by various methods. Best viewed in colour.

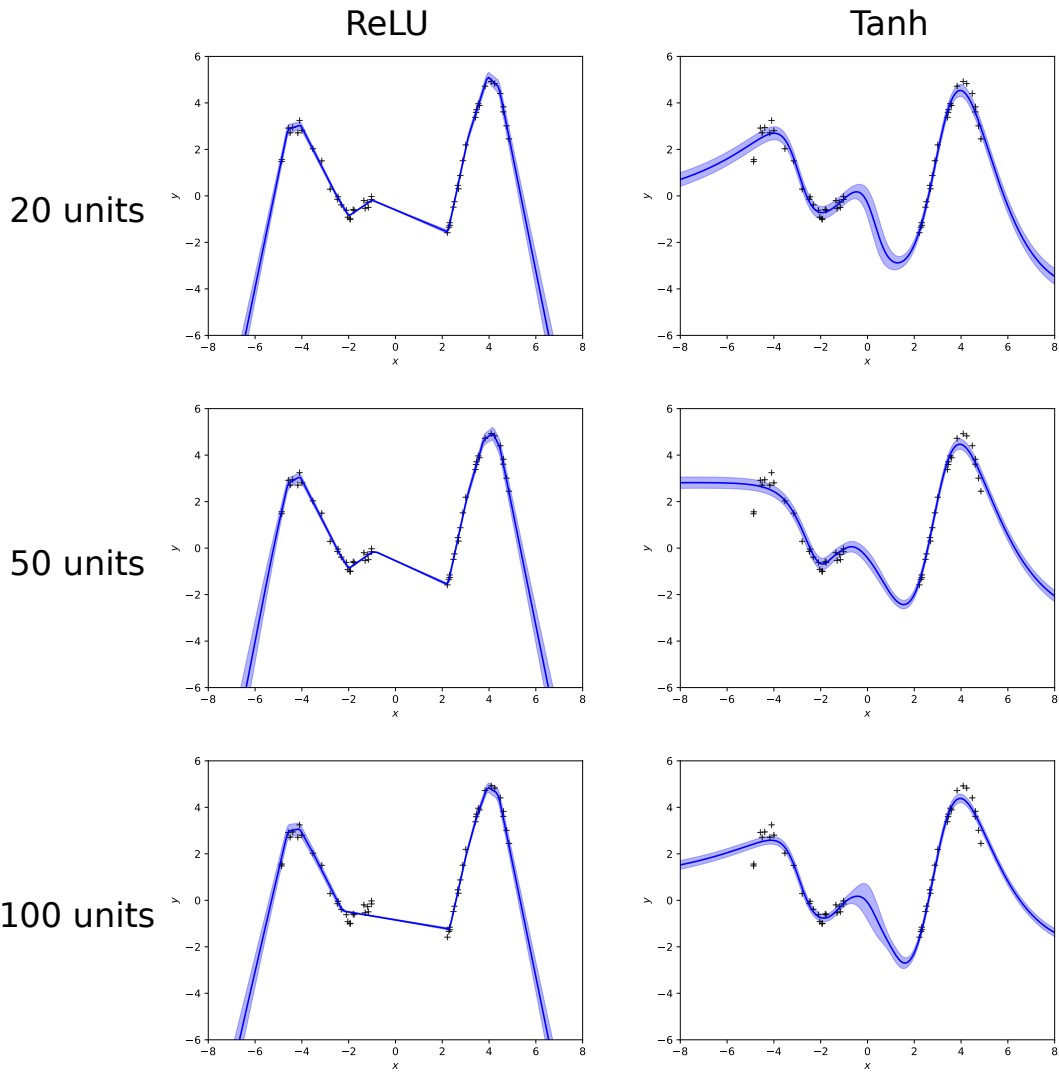

Figure 16: Performance of GP-metarep on a toy regression problem, with various numbers of hidden units and different activation functions. Best viewed in colour.

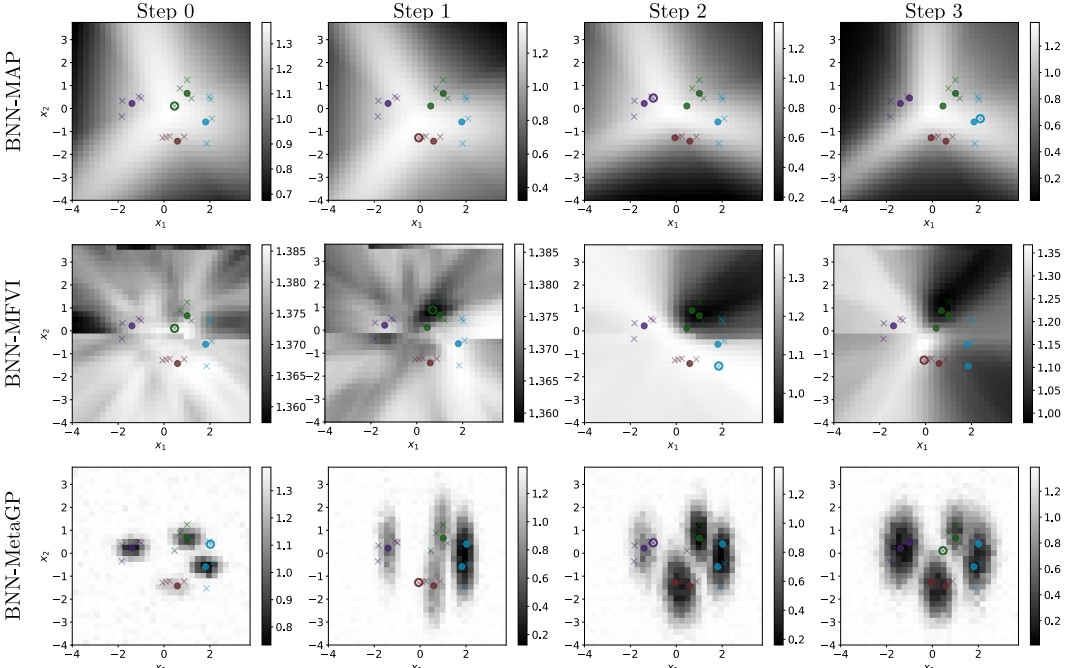

Figure 17: Active learning with BNNs using maximum a posteriori estimation [BNN-MAP], mean-field Gaussian variational inference [BNN-MFVI] and a meta-GP hierarchical prior [BNN-MetaGP] on a toy multi-class classification task. For each plot, the filled circle markers are the current training points, with different colours illustrating different classes. The shaded crosses are the examples in the pool set, one of which we wish to pick and evaluate to be included in the training set. The unfilled circle markers are the examples from the pool set selected at a step. The objective function for selecting points from the pool set is the entropy of the predictive probability. Best viewed in colour.

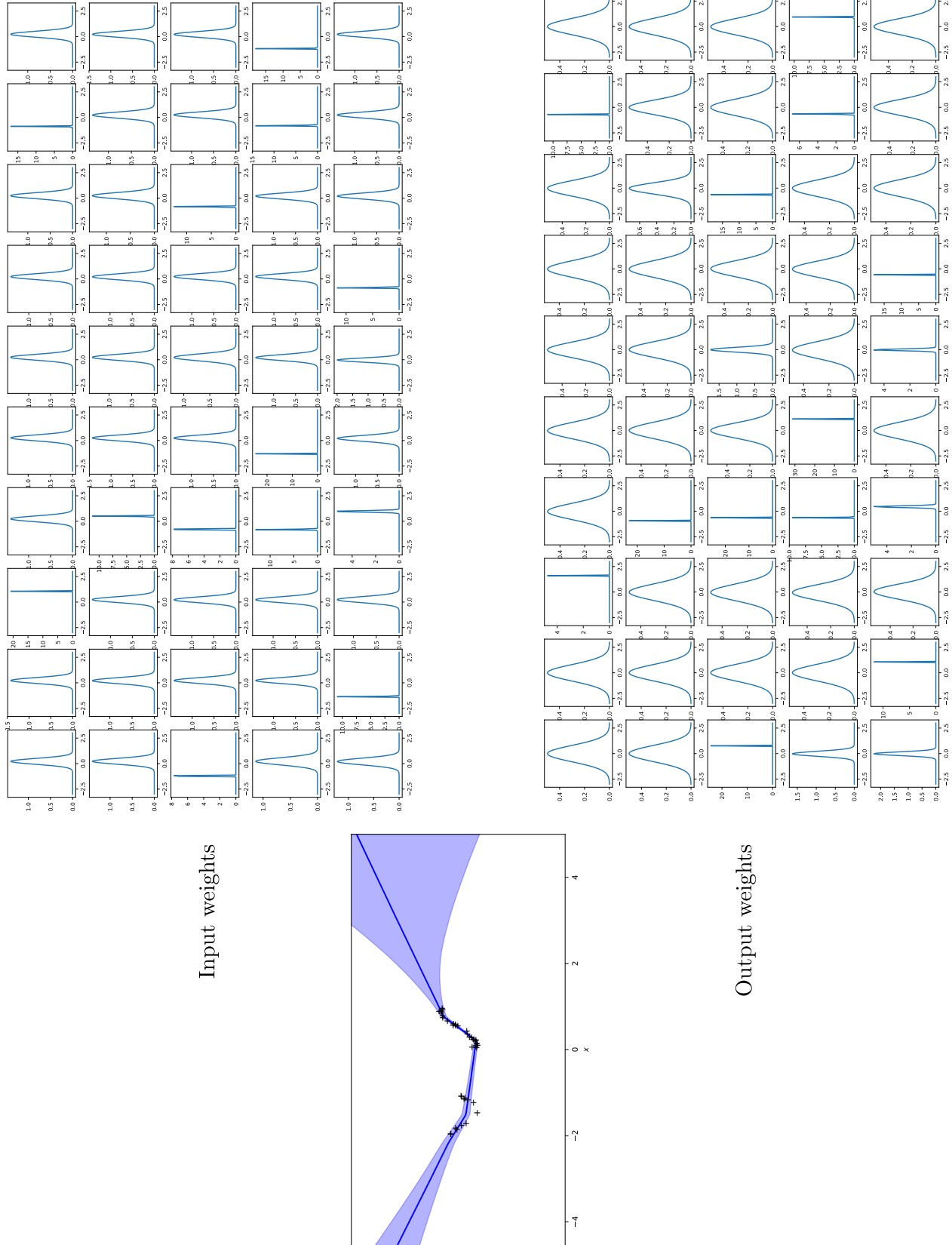

Figure 18: Left: Training data and predictions using a Bayesian neural network with a mean-field variational Gaussian approximation. The network has one hidden layer of 50 rectified linear units, hence there are 50 input weights and 50 output weights. Right: the marginal variational approximations over the weights after training. Most hidden units are pruned out after training, as observed in Trippe & Turner (2018). In this case, there are 12 active hidden units. We include the histogram of the weights in each layer in fig. 3.

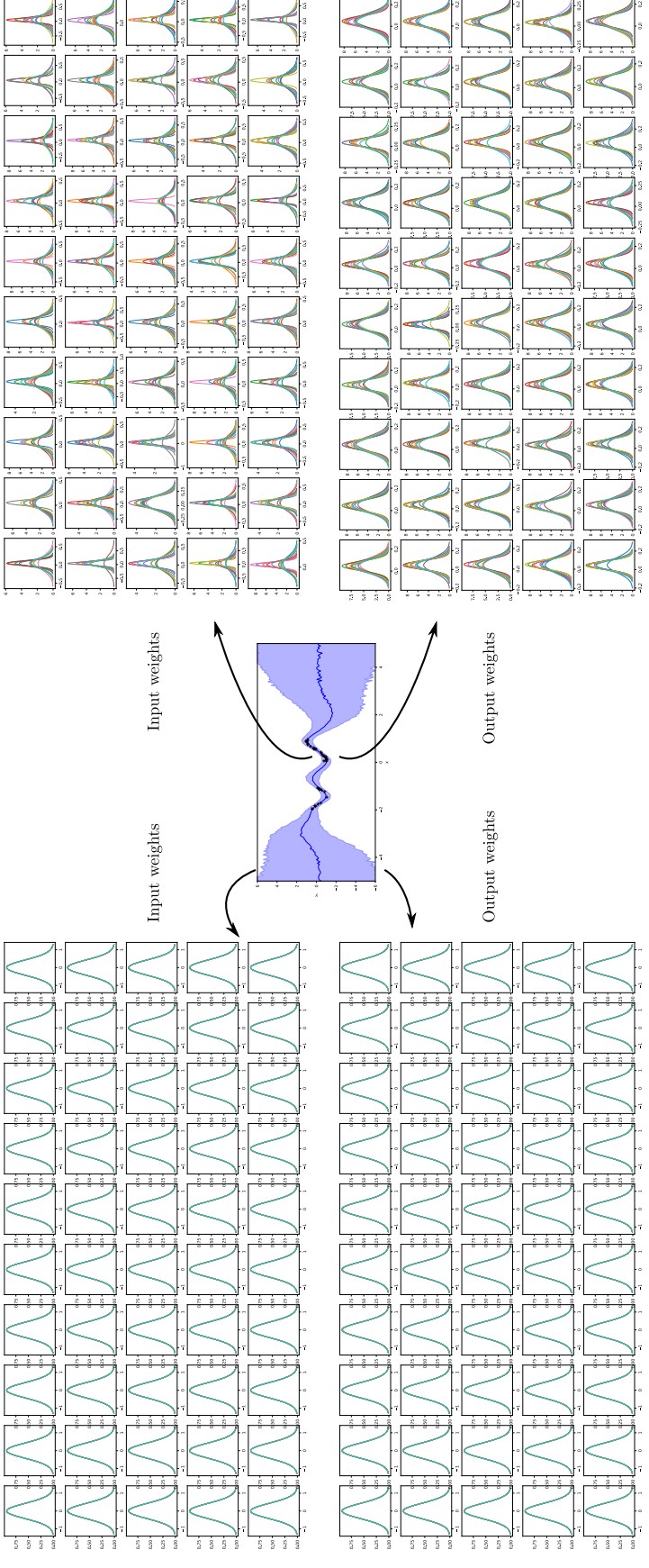

Figure 19: Middle: Training data and predictions using a Bayesian neural network with an input-dependent meta GP prior. Similar to fig. 18, the network has one hidden layer of 50 units. The distributions of the weights in the network, given an input, are included. Note that, for each weight, we first sample the latent variable $z$, and then propagate this sample through the meta GP mapping to obtain a distribution of each weight. In each subplot, we plot 10 distributions corresponding to 10 different $z$ samples. On the left figures, as part of the input to the GP is far away from the training inputs, the weights become uncertain which leads to high uncertainty in the prediction. On the contrary, around the training points, the GP outputs are more certain and more diverse. Note that, the marginal of each weight is an infinite mixture of Gaussian densities. Best viewed in colour.

