# OpenReview forum: "Gaussian Process Meta-Representations Of Neural Networks"
_ICLR.cc/2020/Conference — Reject_

### Official Review · AnonReviewer1 · 2019-10-23
**Official Blind Review #1**

**Rating:** 6

**Review:**

This paper proposes an improvement over Probabilistic meta-representations of neural networks by replacing the NN parameterization of the network weights given latent variables by a probabilistic distribution whose mean is distributed by GP. Inference of the induced hierarchical model is achieved by variational inference and various
approximations when needed.

Comments:

1. The authors claim that the proposed method aims to increase the robustness in the small data settings
and improve its out-of-sample uncertainty estimates. The second part is well justified.
Could the author elaborate further on the high level intuition behind the first part?

2. What exactly is the gain obtained by replacing a NN parameterization with a GP parameterization?
It seems like the proposed method gains the ability to model uncertainty, but potentially incurs performance trade-off
from a series of approximation.

3. Following from comment #2, I am a bit surprised by the lack of comparison against the work of
Karaletsos although this work was built on top of it. I am interested to understand the mentioned
trade-off in practice.

4. Regarding the practical consideration in Eq.(11), I think it kind of defeats the purpose of setting up
the latent variables so that "weights in a layer and across layers are explicitly correlated in the
modelling stage" (Section 2.2). Do all experiments presented in the paper employ this practical
consideration?

5. For the text before Eq. (8) should the inducing inputs be xu instead of zu? It seems like a systematic
typo here because in the formulation for the lower bound it becomes p(u|xu) again instead of p(u|zu)

6. Is Kuu in Eq.(9) computed by taking Kronecker product of K_in, K_out and K(xu,xu) or just K(xu,xu).
I am under the impression that it is the former, in which case taking the inversion is costly (cubic in the number
of latent variables). This would not permit large NN anyway, which kind of defeats the purpose. It explains the choice
of small architecture in your experiments as well.

Overall comment:

I think the paper presents an interesting idea but I have questions regarding its practical significance as highlighted in my specific comments above. I hope the authors would clarify these so I can converge on a final rating.


**Experience Assessment:**

I have read many papers in this area.

**Review Assessment: Checking Correctness Of Derivations And Theory:**

I assessed the sensibility of the derivations and theory.

**Review Assessment: Checking Correctness Of Experiments:**

I assessed the sensibility of the experiments.

**Review Assessment: Thoroughness In Paper Reading:**

I read the paper at least twice and used my best judgement in assessing the paper.

---

> ### Author Response · Authors · 2019-11-12
> **Response to Reviewer 1**
>
> We thank you for your thoughtful review. Please note the general responses we have added. We also address your individual questions in the following:
>
> 1. Re robustness in the small data setting: Thanks for the suggestion. We have included a qualitative analysis of the performance across different number of training points in appendix A.1.
>
> 2. Re gain obtained by using a GP over a NN: The main advantage is to have a Bayesian non-parametric component in the loop that has fewer parameters and for which we can leverage a rich literature on accurate sparse approximation inference, rather than a parametric one with the inference challenge remained largely unsolved. The GP mapping also allows convenient deployment of the input-dependent kernel which shows additional benefits in many practical settings.
>
> Re performance trade-off: In general, sparse GP approximations based on inducing points can match the performance of  exact GPs if a sufficient number of inducing points is used (see e.g. Bui et al (2017), Burt et al (2019)). We also provide an empirical analysis of the diagonal approximation in appendix A.1.
>
> 3. Re compare to Karaletsos et al (2018): Thanks for the suggestion. We have compared to this work in several toy examples in section 6.1 and appendix A.2, and observed that GP-metarep tends to perform similarly to NN-metarep, but Local-GP-metarep has many advantages as shown and actually also converges faster than either GP-Metarep or NN-Metarep. We will provide a more thorough comparison in the next version of the paper.
>
> 4. Re equation 11: We would like to note that this diagonal approximation is conditioned on the latent meta-representation variables z and, as such, the parameters will become correlated when the variables z are integrated out. Indeed, a main advantage of this model is that correlations are captured through the hierarchical dependence on individual Z’s by multiple weights rather than through the output of the mapping.
> In addition, we can use a non-diagonal approximation (see appendix A.1) without hurting the computational complexity. However, we observed that the diagonal approximation performs as well as other structured and potentially more expensive approximations. In light of this analysis, all experiments considered in the paper employ the diagonal approximation.
>
> 5. Re xu vs zu: Thanks for pointing this out. It is indeed a typo. We have fixed this.
>
> 6. Re Kuu in eq 9: It is just K(zu, zu), that is, Kuu is the covariance matrix evaluated at zu. Let M be the number of inducing points then Kuu is a MxM matrix. As with typically done in the sparse GP literature, M is much smaller than the number of weights in the network or the number of data points, which admits tractable computation regardless of the network size.
>
> Thanks again for your review. Please let us know if you have any further comments.
>
> References:
>
> Bui, Yan, and Turner, A Unifying Framework for Gaussian Process Pseudo-Point Approximations using Power Expectation Propagation, JMLR 2017
> Burt, Rasmussen and van der Wilk, Rates of Convergence for Sparse Variational Gaussian Process Regression, ICML 2019

---

### Official Review · AnonReviewer2 · 2019-10-23
**Official Blind Review #2**

**Rating:** 6

**Review:**

**Summary**: This paper proposes a hierarchical Bayesian approach to hyper-networks by placing a Gaussian process prior over the latent representation for each weight. A stochastic variational inference scheme is then proposed to infer the posterior over both the Gaussian process and the weights themselves. Experiments are performed on toy regression, classification, (edit: post rebuttal) and transfer learning tasks, as well as an uncertainty quantification experiment on MNIST.

post rebuttal (noticed recently): Many apologies for updating the review the day before the deadline; however, I recently remembered that Kronecker inference is often used in variational methods - particularly within the vein of literature of deep kernel learning. Indeed, structure exploiting SVI was proposed in Stochastic Variational Deep Kernel Learning, https://arxiv.org/pdf/1611.00336.pdf, and this method is currently the default in Gpytorch: https://github.com/cornellius-gp/gpytorch/tree/master/examples/08_Deep_Kernel_Learning .
Furthermore, Kronecker inference for non-Gaussian likelihoods for Laplace approximations was proposed back in 2015: http://proceedings.mlr.press/v37/flaxman15.pdf.
I am not updating my score because it would be unfair; however, the record should be set somewhat straight here.

post rebuttal: Thank you for the many clarifications and detailed responses. I'm now satisfied with their many changes and and tend to accept this paper despite the experimental results being somewhat limited. I would really encourage the authors to fix the color schemes (please less black and more brighter colors) on their decision boundaries plots however.

**tldr**: While I appreciate the concept of this paper, I tend to reject this paper because I find the experimental results to be on too small scale of datasets. Specifically, I would like to see either a larger scale problem being solved with this kind of approach or a tough to model applied problem that is solved with this approach.

**Originality**: As far as I can tell, this seems to be a novel approach to hyper-networks. Neural processes (Garnelo et al; 2018) propose a somewhat similar approach to training – with a latent process over some stored weight space. However, even that is quite distinct from the method proposed in this paper, and I tend to prefer this approach.

**Quality**: I really appreciate the merging of neural network and Gaussian process methods; however, tragically, I do wonder if the proposed approach combines the worst of both worlds – the necessity of architecture search for neural networks with the choice of kernel function (as illustrated in Figure 5).
If the method is truly kernel dependent, is it also architecture dependent? That is, is it robust to different settings of nonlinearities and depths?

Active learning experiment: While I appreciate the comparison here, it seems like here standard HMC should be trainable over well-designed priors on these architectures. So why not include a comparison instead of just MFVI?

**Significance**: Unfortunately, I think that the experiments section is just a bit too limited to warrant acceptance right now. This is despite the fact that I really do appreciate the thoroughness and thoughtfulness of the experiments as they are.

Specifically, in Section 6.2 why is the metaGP prior only applied to the last layer of the network? If as I suspect, it is due to the complexity and difficulty of inference, that makes the method doubly tough to use in practice. With that being said, to only have experiments on the last layer implies that one should compare to Bayesian logistic regression and linear regression on the last layer of neural networks (e.g Perrone et al, 2018 and Riquelme et al, 2018). Experiments with other methods that combine Gaussian processes with representations on the final layer (e.g. Wilson et al, 2015) are also probably worth running.

Figure 4 is a very well-done experiment, if a bit tough to read. I’d suggest that the out of distribution examples get their own figure, with the in distribution examples going into the appendix. I’d also suggest computing the expected calibration error (Naeini et al, 2015) for in and out of distribution examples on the test sets for both MNIST and K-MNIST in order to have quantitative results on the entire test set.

To recommend acceptance, I’d really have to see experiments on either a CIFAR sized dataset for classification or a larger scale regression experiment. A larger dataset on either transfer learning (after all you do have a meta-representation over functions that the NN can learn), a larger active learning experiment, or semi-supervised learning.

**Clarity**: Overall, the paper is well-written and mostly easy to follow. The meat of the paper is found in Section 4, which I found a bit difficult to follow.

(edit: post rebuttal. This concern is somewhat resolved due to the field not being well developed in this area, although it is a useful place to possibly extend the method in the future.) My primary concern here is that the prior ends up becoming Kronecker structured (after Eq. 7), so it isn’t clear to my why dense matrices and dense variational bounds have to be derived in this setting. Can one not follow the lead of the Gaussian process literature (e.g. Saatci 2012, Wilson & Nickisch, 2015) to exploit the Kronecker structure here to make computation of the log likelihoods fast?
(edit: post rebuttal. This concern is somewhat resolved.) As a result, it’s not immediately clear to me why a diagonal approximation (Eq. 10) is even necessary?
Furthermore, this may be a setting where iterative methods (e.g conjugate gradients and Lanczos decompositions as in Pleiss et al, 2018) for the predictive means and variances may shine and be fast.
I do agree that the approximation in Figure 2 does seem to be relatively accurate, although I would ask the authors to compute a relative error for that plot if possible. Additionally, what is the strange high off diagonal correlations in the marginal covariances?

(edit: post rebuttal. Thank you for the clarifications here.) Finally, I was a bit confused by the effect of adding the input dependent kernel in Section 3; this seems to make the weights much more complicated to model – now each data point has its own set of weights and therefore, we might have to store considerably more weight matrices over time. Could the authors perform a set of experiments showing the necessity of this kernel matrix in the rebuttal?

**Minor Comments**:
-	Above Eq. 9, “splitted” should be split.
-	Figure 3: could the data points be plotted in a brighter fashion? On a dark background, they are quick tough to see. Additionally, what is the difference between the two levels of classification plots?


References:

Naeini, et al. Obtaining Well Calibrated Probabilities by Bayesian Binning, AAAI, 2015. https://www.ncbi.nlm.nih.gov/pmc/articles/PMC4410090/

Perrone, V, et al. Scalable Hyperparameter Transfer Learning, NeurIPS, 2018. http://papers.nips.cc/paper/7917-scalable-hyperparameter-transfer-learning

Pleiss, G, et al. Constant Time Predictive Distributions for Gaussian Processes, ICML, 2018. https://arxiv.org/abs/1803.06058

Riquelme, C, Tucker, G, Snoek, J. Deep Bayesian Bandits Showdown, ICLR, 2018. https://arxiv.org/abs/1802.09127

Saatci, Y. Scalable Inference for Structured Gaussian process models, PhD Thesis, U. of Cambridge, 2011. http://mlg.eng.cam.ac.uk/pub/pdf/Saa11.pdf

Wilson, AG and Nickisch, H. Kernel Interpolation for Scalable Structured Gaussian Processes, ICML, 2015. http://proceedings.mlr.press/v37/wilson15.pdf

Wilson, AG, et al. Deep Kernel Learning, AISTATS, 2015. https://arxiv.org/abs/1511.02222


**Experience Assessment:**

I have published one or two papers in this area.

**Review Assessment: Checking Correctness Of Derivations And Theory:**

I assessed the sensibility of the derivations and theory.

**Review Assessment: Checking Correctness Of Experiments:**

I carefully checked the experiments.

**Review Assessment: Thoroughness In Paper Reading:**

I read the paper thoroughly.

---

> ### Author Response · Authors · 2019-11-12
> **Response to Reviewer 2 Pt. 1/2**
>
> Thank you for your long and detailed review. We have addressed your concerns in both the statement to all reviewers and will address your questions here individually:
>
> Re quality, combining neural networks and GPs: In this work, we are changing the prior over the network parameters, while the underlying mapping from inputs to outputs is still a neural network. This prior, driven by a Gaussian process and judiciously chosen latent variables as well as an optional input-dependent kernel, provides a structured, yet flexible way to encode the modelling assumptions and prior knowledge to the network parameters. It does, however, also facilitate black-box modeling by simply utilizing an RBF or Matern kernel for the kernel on Z (we always use an RBF kernel here) and similarly generic kernels for the input-dependent kernel (we use both and RBF kernel when quantifying uncertainty but also explored periodic kernels when wanting to model temporal/periodic models). One can dig much deeper into the GP/kernel literature to capture more interesting combinations of kernels or perform kernel learning in future work, but we believe as presented the kernels we use are generic and useful and did not require hard manual tuning whatsoever.
> We also note that a big part of the prior lies in the latent variable structure Z, which we take as given, but could potentially be enriched with more prior knowledge. Our contributions are orthogonal to that aspect.
> We believe our model presents clear advantages in terms of representation over other priors or hyper-network architectures which require hard architecture choices and are not interpretable whatsoever.
>
> Re quality, active learning + HMC: we have focussed on approximate inference here. We agree that a comparison to HMC for networks with a Gaussian prior over weights or a Gaussian prior with a hierarchical prior over the prior variance would be interesting. We note that we could also apply HMC for our own model for a fair comparison.
>
> Re significance, last layer: No, scalability when using metaGP over all layers is not an issue, the proposed algorithm can handle that. Please note the diagonal approximation and our result demonstrate its empirically strong performance in Sec. A.1  and that it is cheap to scale to arbitrary networks. We will include such result for all layers for sec 6.2 in the next iteration. In all other experiments, metaGP is applied to all layers.
>
> Re significance, compare to other models: Thanks for the suggestion. We will compare to Bayesian LinReg/LogReg or GP on the last layer. We would like to note that the network architecture remains unchanged in our settings, just the prior on weights is changed, compared to deep kernel learning (Wilson et al, 2015) in which the last layer is replaced by GPs.
>
> Re significance, figure 4: Thanks for the compliment, we indeed tried to provide a thoughtful experiment demonstrating the ability of the model to say “I don’t know” when appropriate. Our result is extremely difficult for Bayesian Neural Networks to achieve, but intuitively what modelers are hoping for when using them. We will compute the calibration error as suggested and make the figures clearer to read.
>
> Re significance, additional experiments: We have added some preliminary experiments on transfer learning and multi-task learning in the appendix. We will expand these in the next iteration. We will also provide a comparison of an HMC network to our local model on a classification task in a next revision.
>
> Re clarity, Kronecker: We would like to note that the literature on Kronecker structure and conjugate-gradient + Lanczos + stochastic trace estimation methods has mostly focussed on GP regression in which the log marginal likelihood, as well as the predictive distribution can be computed analytically. Extending these methods beyond GP regression to models with latent variables and non-Gaussian likelihoods is an active research area and beyond the scope of this paper. In particular, Kronecker-factorized models with inducing points present some additional challenges. In addition, our experiments with fully correlated GPs vs. diagonal approximations revealed that diagonal approximation in our model tend to outperform the fully correlated one, perhaps due to the fact that off-diagonal correlations are marginally captured to a large extent by the latent variables Z and learning becomes easier. The fully correlated model is a superset of the Kronecker-approximated one.
>
> Re clarity, figure 2 and diagonal approximation: These figures show the covariance structures after training using various approximations. We could include figures showing the difference covariance structures with all other parameters fixed. Importantly, the diagonal approximation here is not a model that tries to minimize the error compared to the fully correlated one, but has a different learning trace altogether.

---

> > ### Author Response · Authors · 2019-11-12
> > **Response to Reviewer 2 Pt. 2/2**
> >
> > Re clarity, input dependent kernel and storing weights: No, we *do not* need to store any weight matrices. We only need to store the variational distributions q(z) and q(u) as in the case without the input dependent kernel. The weights for each data train/test input can be computed/sampled *on the fly* at training/prediction time. As such, our local weight model greatly increases the expressivity of the network and facilitates using assumptions such as periodicity when available for the input kernel. We consider the ability to utilize this one of the main reasons to derive the GP prior on weights and a strong supporting element for our paper, as this allows modelers to express inductive biases about data similarly to the way it is done in GP literature (in the trivial case: just to capture uncertainty), while still using weight-based mappings. The results we are able to show for both uncertainty and models of inductive biases, i.e. for periodic functions in the small data setting, speak for themselves.
> >
> > Re clarity, necessity of the input-dependent kernel: We showed the utility of this kernel in various experiments including out-of-distribution uncertainty evaluation, active learning and encoding inductive biases through this kernel. In our multi-task learning experiments (new addition) we demonstrate how the local kernel alone can be varied per task (incurring the low cost of just learning input-kernel parameters per task) while sharing latent variables Z across tasks to allow inferring a model on multiple tasks jointly and can similarly be adapted on the fly to generalize to a new task. This is a novel form of disentangled representation for weights which change per task AND per datapoint and actually works in practice as a straightforward application of our model if we imagine the local kernel to be mapped over a task-plate.
> >
> > Re minor comments: We have fixed the typo and will improve the figures. The background color of the classification plots show the predictive entropy.
> >
> > We also thank for the references, we are aware of them and will add them appropriately in a discussion.
> >
> > Thanks again for your review. Please let us know if you have any further comments.

---

> > > ### Comment · AnonReviewer2 · 2019-11-12
> > > **Thank you for the thoughtful response (part 2)**
> > >
> > > 9) [input dependent kernel and storing weights] Thanks for the clarification here, I'd encourage further revision of the writing to make the fact that the sampling can be done on the fly with minimal memory overhead.
> > >
> > > 10) [necessity of the input dependent kernel] Thank you for including the transfer learning experiments. I think that they are a welcome and strong experiment for the paper.
> > >
> > > Minor: Figure 9 now seems to be overly zoomed in and part of the legend is cut off. I'm assuming that this will be corrected in the next iteration.

---

> > ### Comment · AnonReviewer2 · 2019-11-12
> > **Thank you for the thoughtful response (part 1)**
> >
> > Thank you for taking the time to respond in depth to all of my questions.
> > I'm going to start numbering comments/responses to be keeping track of everything.
> >
> > 1) [combining neural networks and GPs] Thanks for the response about dependence on latent variables + Gaussian process kernels. However, I'm still a bit worried about the potential dependence on architecture - the scenario that I'm attempting to parse out is really what makes this model work properly. Is it the architecture of the neural network or the prior function? This concern is based on the somewhat worrying performance of the RBF kernel as compared to the periodic kernel in Figure 5.
> >
> > 2) [active learning + HMC] Indeed this experiment would be interesting, as would a fully sampling based approach for your model. What is the drawback from implementing it here? I can't imagine that the likelihood function and its gradients are difficult in these small data situations.
> >
> > 3-6) I will update on seeing these figures.
> >
> > 7) [Kronecker] Thanks for the clarifications throughout here. I'm satisfied that this could be an open research question in and of itself.
> >
> > 8) [diagonal] Thanks again for the clarification.
> >
> > Minor additional comment: Could the title lines in Figure 5 be moved to a separate table and removed from the figure itself? They are quite hard to read without zooming in completely.

---

> > > ### Author Response · Authors · 2019-11-13
> > > **Response to point (1)**
> > >
> > > Dear Reviewer #2,
> > >
> > > First of all we highly appreciate your fast response and the resulting opportunity to address your concerns.
> > >
> > > We are working on further experiments as per your request after having updated the paper once already. In the meantime, we wanted to make sure we follow your comment (1) here and expand on our current understanding of it:
> > >
> > > With Regards to Architecture:
> > > We provide a brief refresher of the underlying model structure for MetaRepresentations that we inherited from the work of Karaletsos et al 2018.
> > > in this unit-centric model, for each node/unit of a neural network a hierarchical latent variable Z_unit is sampled from a prior. Then, for each weight connecting a unit A and a unit B, these latent variables are used as inputs to a mapping function to propose a  weight distribution.
> > > This affords the model to propose weight distributions independent of architecture, as units are the building blocks of all neural network architectures and the model is following the structure of the given architectures by explicitly learning Z_unit for each unit, observed or hidden.
> > > Our Gaussian Process mapping then only has to perform one-dimensional regression to each weight and as such has no limitations in terms of architectures or connectivity patterns of weights.
> > > Weights are sampled on the fly conditioned on the latent variables Z and rich correlations are modeled by virtue of the weight connectivity patterns of each node.
> > > For example, if the latent variable for an input node changes, all weights connected to that input feature will change in a correlated fashion.
> > > If, similarly, a hidden node changes, all weights connected to the nodes in all connected layers will change accordingly.
> > >
> > > In our correlation plots in Figure 2 these structures are visible as the parallel off-diagonal bands which are the hidden units tying together the weights. These are not blocks, since the model we are showing only has a single input and output unit.
> > >
> > > Crucially, learning of weights is superseded in this representation by learning of the latent variables Z, which are significantly smaller in number as they describe the backbone of the neural network and not the weight matrix which roughly scales in quadratic order in terms of size compared to the units.
> > > Indeed, our variational inference scheme in Sec. 4 reveals precisely how the weight terms cancel out for inference and learning and are only stochastically sampled from the approximate posterior for ELBO evaluation. Weights then can be sampled on the fly given the posterior on Z and the mapping.
> > >
> > > For further clarifications we suggest our overview in Sec. 2.2 or indeed the paper itself. We hope this clarifies that our prior is generic for any architecture comprised of units which are connected by weights.
> > >
> > > With regards to Figure 5:
> > > We believe that Figure 5 shows that the MetaGP model without the local kernel behaves comparably to the NN-Metarep model and only fits the visible data and extrapolates accordingly.
> > > However, the local kernels afford the model with interesting additional tools for generalization:
> > > 1. when using local kernels with inducing points, we also learn inducing points for the inputs (or transformed inputs epsilon when the inputs are high dimensional). During application of the model, this allows the weight prior with local kernels to adapt weights per data-point by comparing new inputs to the inducing inputs, thus being able to modulate weight priors for inputs that are far away from training data accordingly. The RBF kernel reveals that the model will increase uncertainty and reverts to an uninformative weight prior as it moves away from the few observed datapoints, which we consider a strong result as evidenced quantitatively at the top of the figure showing that the RBF local kernel model achieves better test error than a non-local model.
> > > Meaning: this model extrapolates and interpolates more usefully than a non-local model, both qualitatively and quantitatively.
> > >
> > > 2. If we use a more adequate periodic kernel as the input kernel, our model can represent function spaces more concentrated around the true function and does even better for interpolation and extrapolation, as the plot and the performance metrics show, since the function space that the weights have to represent is regularized more tightly through our weight prior local-GP-Metarep.
> > >
> > > We hope that, together with the MNIST uncertainty results, this clarifies that we can get useful local models utilizing generic Black Box kernels like the RBF kernel, but if available or wanted can also encode prior knowledge for functional regularization such as periodicity. This is not a limitation, but an extra tool in the modeler's toolbox with highly appealing results compared to generic weight priors which cannot generalize in this way.
> > >
> > > In case we did not satisfy your questions, could you please further clarify your points on architecture and why you consider the RBF kernel behavior worrying?

---

> > > > ### Comment · AnonReviewer2 · 2019-11-14
> > > > **Thank you for the clarification of Figure 5**
> > > >
> > > > Thanks for your response.
> > > >
> > > > 1) I should probably be a bit more explicit - what is the performance of your model like if you fix the GP prior and vary the architecture of the neural network (e.g. number of hidden units or number of layers)?
> > > >
> > > > Thank you for clarifying that the GP prior itself only has to perform a one d regression on the hidden units Z mapping that to the output weights. Now that I better understand this, I can see that the approach ought not to be too slow.
> > > >
> > > > Very specifically, I'm concerned that there are failing cases in the extra degrees of flexibility of your model due to a practictioner having to choose a) number of hidden units, b) nonlinearity, c) depth d) prior kernel, e) number of inducing points.
> > > >
> > > > 2) With respect to the results presented in Figure 5, I think that you've clarified these points to my satisfaction. I'd again suggest emphasizing these arguments in the final version.

---

> > > > > ### Author Response · Authors · 2019-11-15
> > > > > **clarification to point 1**
> > > > >
> > > > > 1) As most of the work for representing weights is performed by the latent variables per unit z_u, varying architectures does not burden the GP part of the prior much.
> > > > > We typically set the number of inducing points to a small amount, e.g. 50, pick an RBF kernel by default for latent variables Z (as explained in other responses) and optimize the kernel parameters according to our loss.
> > > > >
> > > > > Inducing point kernels can represent the spectrum of weights of different architectures pretty well even for deeper networks, as weights empirically mainly have differing spectra in input and output layers, but quite narrow dynamic ranges across hidden layers.
> > > > > As such, we can represent deeper and wider networks without varying the kernel design, as long as the corresponding latent variables Z are jointly inferred.
> > > > > We have not observed the necessity for variations of the K_z kernel function or choices with respect to number of hidden units, architecture or nonlinearities used in the network.
> > > > >
> > > > > We note that performance of the model degrades if too few inducing points are used (for instance 10), as is to be expected, since the function space of the GP becomes too restricted.
> > > > >
> > > > > We will add a discussion and some results to the paper to make the reader understand:
> > > > > ->the role of # of inducing points (has to be large enough to represent the spectrum of weights necessary per task, but complexity does not really scale with network size)
> > > > > ->the effect of changing architecture width/depth to demonstrate that the GP-prior part of the model will handle this fine (i.e. by changing networks to be 20unit, 50unit or 20-20 unit architectures in a fixed task).
> > > > >
> > > > > The reviewer brings up a very interesting general question here:
> > > > > Which weight priors or weight prior properties are paired well with particular nonlinearities in combination with architectures and datasets? Can we explain why? Can we possibly learn such optimal pairings?
> > > > > While outside the scope of our paper, we are unaware of significant literature studying this topic with the possible exception of the burgeoning literature on architecture search yielding some related empirical findings and find it appealing to explore this topic more in the future.

---

> ### Author Response · Authors · 2019-11-15
> **Please note our second revision adding experiments you have suggested**
>
> Dear Reviewer #2,
>
> We have added material to the paper as previously stated.
>
> In particular, per your suggestions:
> 1. we compare (favorably) to Deep Kernel Learning in our second revision and show this with an uncertainty evaluation experiment.
> 2. we ran the model for some variations of neural networks with different sizes and nonlinearities without varying our kernel choices.  We hope this addresses your architecture concerns that we have discussed.
>
> Please note our second general comments to reviewers and thank you for interacting with us during the rebuttal period to clarify your questions . We believe the resulting experimental evidence and discussions provide more clarity for all readers in support of our work.

---

### Official Review · AnonReviewer3 · 2019-10-25
**Official Blind Review #3**

**Rating:** 6

**Review:**

The paper presents two models extended from a meta-presentation of neural networks (Karaletsos et al. 2018) in which neural network weights are constructed hierarchically from latent variables associated with each layer. The mappings from latent space to weight space are used Gaussian Processes instead of neural networks.

The proposed models include (1) MetaGP which directly replaces neural network assumption by Gaussian process prior and (2) MetaGP with contextual information which further takes into account input information via performing the multiplication between the kernel function over latent space and kernel function over input space.

Variational inference is followed by the pseudo-inducing point approach.

Experiments are conducted in both toy data sets and benchmark data sets, demonstrating several points i.e. uncertainty quantification, inductive bias.

Pros:
The paper is clearly written.
Introducing inductive bias or functional regularization for neural network
Interesting capability of measuring the uncertainty of out-of-sample data. This can be one of the reasons that the method performed well in active learning.

Cons:
The approach is incremental or not-so-novel in terms of meta-representation for neural networks.

Comments and questions:
The prior distribution for latent variable $z$ is not specified. I assume the prior is independent Gaussian.
$z$ is unknown beforehand. How are inducing locations for latent variable $z$ initialized?
 Do you think that there is a connection between contextual metaGP and residual nets? Can skip connections from the input layer to certain layers be considered to be similar to the idea of incorporate input kernel in the paper?
Can you comment on the convergence of the estimation of the last term in the variational bound? The MCMC takes two stages of stochasticity: (1) sample $z_k$ and $V_k$ and (2) then estimate $F_k$ using reparameterization. This can make the convergence slow (https://arxiv.org/abs/1709.06181).
Minor: a missing period in Sec. 6.3 “quadratic kernel In this example”

**Experience Assessment:**

I have published in this field for several years.

**Review Assessment: Checking Correctness Of Derivations And Theory:**

I assessed the sensibility of the derivations and theory.

**Review Assessment: Checking Correctness Of Experiments:**

I assessed the sensibility of the experiments.

**Review Assessment: Thoroughness In Paper Reading:**

I read the paper at least twice and used my best judgement in assessing the paper.

---

> ### Author Response · Authors · 2019-11-12
> **Response to Reviewer 3**
>
> Thank you for your thoughtful review, we would ask you to kindly take the general comments into account and will also respond individually to your comments in the following:
>
> Re incremental contribution: we would like to note that the proposed model and method allows the prior over the network parameters that can be flexibly modelled by a GP and can be further enriched by the input-dependent kernel. We demonstrate the utility of the proposed approach on a suite of tasks showing accurate prediction, rich inductive biases via kernels and calibrated predictive uncertainty. This principled merger of the worlds of Neural Networks and Gaussian Processes has to our knowledge not been executed as performed here and clearly demonstrates empirical value in addition to conceptual elegance.
> Re prior over the latent variable: yes, we use a diagonal Gaussian prior, as typically done in many other latent variable models. This is noted in our paper in Sec.2.2 above Eq. 2.
> Re inducing point initialisation: we randomly initialize the inducing inputs and make sure that these initial values are similar in value to the initial mean of the latent variable z. Note that this init strategy is also used in the GP latent variable model (Lawrence 2006, Titsias and Lawrence 2010).
> Re connection to residual nets: Thanks for the suggestion. We agree this connection is a very appealing idea to investigate in future work. We would like to note one clear difference: skip connections directly add the activations of a layer to the output-features of the next layer while in our case, inputs are used to adapt the weight prior only.
>
> Re convergence: we observed that the proposed inference scheme converges after a reasonable number of epochs, comparable to that of the standard mean-field Gaussian variational inference. We will include the full training curves in the next iteration.
>
> Re missing period: Thanks, we have fixed this.
>
> Thanks again for your review. Please let us know if you have any further comments.

---

### Author Response · Authors · 2019-11-12
**General Response To Reviewers**

Dear reviewers,

Many thanks for your detailed and constructive feedback.

We briefly recap the contributions in our paper:

1. We introduce a neural network weight prior based on unit-variables (pre-existing work) utilizing a GP-hypernetwork (novel work). This allows us to derive hierarchical weight-models without the need for neural networks in the loop by utilizing elegant Bayesian non-parametric models with very few parameters.

2. A great advantage of the novel GP-based weight prior is that it sets the foundation to allow us to represent the weight kernel as a product kernel of the latent-variable kernel (K_z) and an Input-dependent kernel (K_x) which  facilitates deriving elegant and scalable per-datapoint weight priors for neural networks and performing functional regularization, thus encoding prior knowledge into the input-kernel similar to the GP literature.

3. We present an Approximate inference scheme for the joint model which facilitates efficient, scalable and tractable inference while jointly inferring variational distributions for the latent variables Z, the inducing variables u and optimizing kernel parameters. Crucially, we obtain scalability to arbitrary networks by a diagonal approximation which maintains the ability to capture correlations induced by the hierarchical latent variables Z, but omits off-diagonal terms in the conditional weight model. In practice we observe that omitting these terms does not hurt performance (see Sec. A.1 in the appendix), as most correlations in the model are captured hierarchically by Z instead of by the GP weight observation model.

4. We present a suite of experiments demonstrating:
-That the GP-MetaPrior model works as intended,
-That the local weight model is greatly beneficial for modeling weight uncertainty when using a generic RBF kernel, in particular when applied to tasks where the model should say “I don’t know”
-That the local weight model also allows utiliziation of kernels expressing prior knowledge, for instance periodic kernels, facilitating a novel form of function space regularization that allows the neural network to express periodic functions and both interpolate and extrapolate accordingly with little training data in settings previous neural network weight priors could not capture.

We believe this work proposes an entirely novel way to combine the benefits of the GP-literature and the benefits of neural networks and empirically demonstrates the value of doing so.
Our work can be understood as an explicit form of functional regularization for neural networks through the weight prior. Our goal in this paper is to introduce this new machinery to the community for modeling input-dependent priors for neural networks and we believe ICLR is an appropriate venue as we propose novel, demonstrably useful and scalable means to learn compact forms of weight representations with clean and principled machinery.

We updated the submission with fixed typos, tightened text, and a suite of experiments that we believe have addressed your concerns and strengthened the paper.

In particular, we include an empirical analysis comparing the diagonal approximation in equation (11) to alternative approximations (e.g. diagonal + low rank or fully correlated) in appendix A.1, a comparison of various methods in the low-data regime in appendix A.2, an investigation of the input-dependent kernel and its applicability to transfer learning (fine-tuning) and multi-task learning/fast adaptation in appendix A.3.
We also intend to take a second pass at the paper before the end of the discussion period to strengthen and clarify the work as necessary.

We will now address each of your reviews individually.

---

### Author Response · Authors · 2019-11-15
**Second General Response and Revision To Paper**

Dear Reviewers,

Based on your feedback we have uploaded a second revision of our paper to include more experiments.

In particular, in this second revision we add:

1. Results comparing Deep Kernel Learning to our non-local and local weight models on the MNIST uncertainty quantification task and other datasets. Our results in supplement A.4 show that our model, while remaining a weight-based model and only modifying the weight prior, outperforms Deep Kernel learning for uncertainty quantification. Please see the corresponding Figures 12, 13, 14 and 15 in the appendix, which show both quantitative evidence as well as examples for particular images as before.

2. We append a result running different sizes and architectures and different nonlinearities with our model to demonstrate the flexibility of the weight prior to such changes. Please note Figure `6 in the supplement A.5 which compares ReLU and Tanh nonlinearities for models with 20, 50 and `100 units. The kernels on Z are identically chosen each time.

Together with our previous additions of experiments adding Multi-task learning, comparing diagonal GP approximations to other choices for the GP and the experiments studying performance in low-data settings we hope we have conclusively addressed the concerns our reviewers have raised.

Along with the discussions and clarifications during the rebuttal period we hope this demonstrated the correctness and empirical value  of our method.

We thank the reviewers for their time and feedback and hope they will consider our updates in their assessments.

---

### Decision · Program_Chairs · 2019-12-19

**Decision:**

Reject

**Comment:**

The authors propose an approach to Bayesian deep learning, by representing neural network weights as latent variables mapped through a Kronecker factored Gaussian process. The ideas have merit and are well-motivated. Reviewers were primarily concerned by the experimental validation, and lack of discussion and comparisons with related work. After the rebuttal, reviewers still expressed concern regarding both points, with no reviewer championing the work.

One reviewer writes: "I have read the authors' rebuttal. I still have reservation regarding the gain of a GP over an NN in my original review and I do not think the authors have addressed this very convincingly -- while I agree that in general, sparse GP can match the performance of GP with a sufficiently large number of inducing inputs, the proposed method also incurs extra approximations so arguing for the advantage of the proposed method in term of the accurate approximate inference of sparse GP seems problematic."

Another reviewer points out that the comment in the author rebuttal about Kronecker factored methods (Saatci, 2011) for non-Gaussian likelihoods and with variational inference being an open question is not accurate: SV-DKL (https://arxiv.org/abs/1611.00336) and other approaches (http://proceedings.mlr.press/v37/flaxman15.pdf) were specifically designed to address this question, and are implemented in popular packages. Moreover, there is highly relevant additional work on latent variable representations for neural network weights, inducing priors on p(w) through p(z), which is not discussed or compared against (https://arxiv.org/abs/1811.07006, https://arxiv.org/abs/1907.07504). The revision only includes a minor consideration of DKL in the appendix.

While the ideas in the paper are promising, and the generally thoughtful exchanges were appreciated, there is clearly related work that should be discussed in the main text, with appropriate comparisons. With reviewers expressing additional reservations after rebuttal, and the lack of a clear champion, the paper would benefit from significant revisions in these directions.

Note: In the text, it says:
"However, obtaining p(w|D) and p(D) exactly is intractable when N is large or when the network is large and as such, approximation methods are often required."
One cannot exactly obtain p(D), or the predictive distribution, regardless of N or the size of the network; exact inference is intractable because the relevant integrals cannot be expressed in closed form, since the parameters are mapped through non-linearities, in addition to typically non-Gaussian likelihoods.

---

> ### Author Response · Authors · 2019-12-20
>
> Here, we would like to strongly express our concerns with the final decision and the rationale listed in the decision comment:
>
> 1. ) Regarding related Kronecker factored inference:
> The model we proposed here uses a product kernel, but is different to standard GP regression and classification models with Kronecker-structured covariance functions in that it has latent variable structure. As we have pointed out to R2, extending Kronecker-GPs to work with GP latent variable models (which has latent variables and GP mappings) like the model we considered here has not been looked at in the literature, including the papers mentioned by the R2 and AC which do not fit our setting.  To clarify: there is no content in the papers mentioned by R2 and AC showing that these methods can be used for a GPLVM-type model with Kronecker structure.
> Moreover, even if there was such evidence, we provide experimental evidence that our diagonal approximation is sound in Appendix A1 as compared to a full GP, as our model introduces correlations by hierarchy and does not require the full covariance GP mapping. As such it does not even hurt our model.
> Consequently, the concerns about the lack of comparison to/use of these Kronecker methods is not relevant/factually wrong and beyond the scope of our submission. We are surprised these clearly inappropriate arguments are still used to reject our submission after explaining the same during the rebuttal period.
>
>
> 2.) Regarding related work on latent variable representations:
> We highly appreciate the pointer to relevant work, but it was brought up with the decision by the AC and not by the reviewers so we could not respond in time to them. The first paper by Pradier et al (2018) is a relevant citation but requires pretraining of an auto-encoder and has not been published in a peer-reviewed conference. The second paper by Izmailov et al (2019) is orthogonal (which requires pre-training of neural networks and does not introduce a prior on networks), and was published and available only two months before the ICLR deadline. Both of these works do not consider input-dependent priors for network weights which is a major contribution of our paper. We would understand the request to cite them, but faulting us for a lack of comparison seems far-fetched as they are not directly comparable.
>
>
> 3.) Regarding a comparison to deep kernel learning (DKL) as pointed out by the AC:
> As requested by R2, we compared to DKL during the rebuttal period and updated the submission accordingly. The results showed conclusively that our model significantly outperforms DKL in a suite of tasks. We would be happy to move the results to the main paper if that were requested by the AC, but are puzzled that this comparison is called ‘minor’ and its existence is not acknowledged as we performed it on our marquee experiment which was praised for rigor by R2.  In addition, DKL is the strongest baseline of all the work pointed out by the AC, so we find the argument that we fail to compare adequately to the literature quite surprising.
>
>
> 4.) Reviewer not convinced by merging of GP and NN:
> We presented technical and empirical evidence that not only is this doable but also that in the case of input-dependent weights it provides the ability to a model to express calibrated uncertainty for interpolation and extrapolation and to be imbued with inductive biases for extrapolation beyond any current network priors.
> At last week's NeurIPS conference a major topic at the Bayesian Deep Learning and Approximate Inference community were precisely these aspects of BNNs for which the manifold talks had no solution and yet we are faced with a decision which undermines publication of a method that directly tackles these issues constructively with a new idea because somebody is not convinced they 'like it' without referring to a clear technical or empirical problem with the work. We are frankly shocked and disappointed by this.
>
>
> We are disappointed by the outcome of this reviewing process.
> We successfully performed our best to convince reviewers to uniformly provide scores above acceptance level by repeated interventions and updates to the paper during the tight rebuttal period based on hard work and interaction.
>
> Observing an outcome with a weak and technically unfounded final assessment is utterly disappointing and undermines the spirit of the rebuttal process. We also feel ICLR missed an opportunity to support novel work with fresh ideas that empirically and technically are sound and tackle existing unsolved challenges in neural networks without providing a technical reason to do so that would help us improve the work.